 # TarGATE: Target-Aware Data Selection via Token-Attenuation Gates

**Xiandi Luo** [* 1]  **Shiwei Li** [* 2 1]  **Haozhao Wang** [1]  **Yihao Ouyang** [1]  **Zhuoqi Hu** [1]  **Yichen Li** [1]  **Xiao Yang** [3]
**Huning Liu** [4]  **Ruixuan Li** [1]

## Abstract

Targeted instruction tuning requires selecting pertinent samples from massive mixed *candidate datasets* guided by a small *reference dataset* reflecting the desired capability. However, efficiently identifying high-quality data amidst noise remains challenging. To address this, we propose **Tar**get-aware **GATE**s (**TarGATE**), a simple yet effective data selection framework that leverages the model's inherent data understanding ability. These gates compute a token-level Information Retention Ratio (**IRR**) to attenuate the output of the feed-forward network, where the instance-level average IRR serves as a quantitative metric for data quality. To align gates' preferences with the target task, we employ a joint optimization strategy utilizing the reference dataset and a subset of candidate data, which encourages the gates to assign higher IRRs to reference-aligned data while suppressing low-quality samples. Extensive experiments across noisy and real-world scenarios demonstrate that TarGATE outperforms related baselines. Furthermore, TarGATE exhibits superior computational efficiency and strong cross-model transferability, enabling smaller selector to effectively curate high-quality fine-tuning data for larger foundation models. The code is available here.

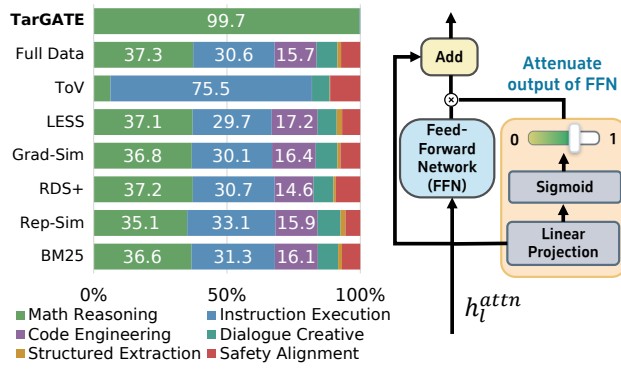

(a) Data distribution    (b) Architecture of TarGATE

*Figure 1.* (a) The percentage of data selected on Tulu-3 (Lambert et al., 2024) by each method using GSM8K (Cobbe et al., 2021) as the target task, with categories labeled by GLM-4.7 (Z.AI, 2025). Mathematical reasoning data makes up the majority of the subset selected by TarGATE. (b) Illustration of the TarGATE structure.

## 1. Introduction

Large Language Models (LLMs) have recently demonstrated remarkable performance across a wide range of complex tasks (Vaswani et al., 2017; Yang et al., 2025; DeepSeek-AI, 2025), a success largely attributed to the critical role of Supervised Fine-Tuning (SFT). By aligning pre-trained models with human intent, SFT serves as the bridge that empowers these models to understand and execute specific instructions (Ouyang et al., 2022; Wei et al., 2022; Gururangan et al., 2020). To support such versatile capability adaptation and ensure broad generalization, the prevailing paradigm typically involves fine-tuning on massive, mixed pools of instruction data that aggregate knowledge from diverse domains (Ivison et al., 2025; Li et al., 2024a).

However, recent research indicates that fine-tuning on massive mixed datasets is not always optimal (Li et al., 2024b; Dohmatob et al., 2025; Sachdeva et al., 2024), while models even perform better on specific tasks when fine-tuned on a carefully selected subset of high-quality data rather than the full dataset (Zhou et al., 2023; Chen et al., 2024; Liu et al., 2024). Nevertheless, identifying such a specialized subset presents a significant challenge, as high-quality data directly

---

[*]Equal contribution  [1]School of Computer Science and Technology, Huazhong University of Science and Technology, Wuhan, Hubei, China [2]Tongyi Lab, Alibaba Group, Hangzhou, China [3]School of Earth Science and Engineering, Hebei University of Engineering, Handan, Hebei, China [4]Department of Computer Science and Technology, Jiangxi University of Traditional Chinese Medicine, Nanchang, Jiangxi, China. Correspondence to: Haozhao Wang <hz_wang@hust.edu.cn>.

*Proceedings of the 43[rd] International Conference on Machine Learning*, Seoul, South Korea. PMLR 306, 2026. Copyright 2026 by the author(s).

aligned with specific target capabilities is often scarce and submerged within noisy, heterogeneous corpora. This challenge leads to the concept of *targeted instruction tuning* (Xia et al., 2024; Zhang et al., 2025; Nikdan et al., 2025; Yu et al., 2024): given a massive, mixed candidate dataset and a small reference dataset reflecting the desired capability, how can we efficiently select the high-quality data most relevant to our target?

In this paper, we aim to measure data quality by utilizing the inherent token-understanding ability of advanced language models. Specifically, we introduce a token attenuation gate within each Transformer block of the language model to attenuate the output of the feed-forward network (FFN). As illustrated in Figure 1(b), the gate is a single linear projection followed by a sigmoid function to produce a scalar activation for each token, which we term the Information Retention Ratio (**IRR**). The IRR allows us to quantify the distinction between meaningless noise and meaningful information within a token. Consequently, the overall IRR serves as a metric for the data quality.

To empower the gates to filter out useless information, we fine-tune the gate parameters using task-related loss while keeping the foundation model parameters frozen. The training data is randomly sampled from the candidate dataset, therefore its quality varies significantly. To enhance the gates' ability to prune irrelevant information, we apply regularization to this data, encouraging a higher rate of token attenuation. However, at this stage, the gates can only distinguish generally low-quality data; they lack an understanding of relevance to the specific target task. To enhance *target awareness*, we incorporate the reference set, which reflects the characteristics of the target distribution (Xia et al., 2024), into the training process. Specifically, we apply a contrasting regularization to the reference data that encourages higher token activation ratios. This aligns the gates' perception of *high-quality data* with the distribution of the target task.

Once the gates are trained, we score each candidate data point by performing a single forward pass through the selector. Then, we calculate the mean IRR, averaged across all tokens and layers, as a quality score to reflect the data's relevance to the target task. Finally, we select the top-$K$ examples to form our fine-tuning dataset. Notably, the model structure or size used for the subsequent fine-tuning can be flexible and is not directly tied to the data selector.

This framework is called **TarGATE** (**Tar**get-aware **GATE**s), and our main contributions are summarized as follows:

- **TarGATE** is proposed, a simple framework where a token attenuation gate is inserted to attenuate the output of the feed-forward network based on a computed Information Retention Ratio (IRR). The average IRR

is directly utilized as the quality score to select data.

- A joint optimization strategy is devised to align the gates with the target task. The gates are trained to assign higher IRRs to reference-aligned data while suppressing randomly sampled candidate data, enabling the precise identification of high-quality samples.

- Comprehensive experiments are conducted across noisy and real-world scenarios, demonstrating that TarGATE consistently outperforms SOTA baselines. Furthermore, superior computational efficiency is achieved through single-pass selection, and strong cross-model transferability is exhibited, enabling smaller selectors to curate data for larger models.

## 2. Related Work

### 2.1. Data Selection in Large Language Models

Selecting high-quality training data is fundamental for the efficient learning and alignment of LLMs (Li et al., 2024b; Sachdeva et al., 2024; Liu et al., 2024). Traditional approaches often rely on heuristic filtering, such as deduplication and lexical overlap metrics like BM25 (Li et al., 2024a; Trotman et al., 2014; Robertson & Zaragoza, 2009), or semantic rating mechanisms (Wettig et al., 2024). Beyond heuristics, representation-based methods utilize embedding similarity, such as Rep-Sim and its variants, to align training data with target distributions (Pezeshkpour et al., 2021; Ivison et al., 2025). Other research frames data selection as a coreset problem, employing clustering or sensitivity sampling to identify representative subsets (Phillips, 2016; Mirzasoleiman et al., 2020; Axiotis et al., 2024). Recent studies have empirically demonstrated that pruning datasets based on difficulty or quality scores can alter neural scaling laws, allowing models to achieve superior performance with significantly less data (Sorscher et al., 2022; Chen et al., 2024; Dohmatob et al., 2025). This "less is more" paradigm is further evidenced by recent reasoning-focused models like LIMO (Ye et al., 2025) and S1 (Muennighoff et al., 2025), which underscore that a curated set of challenging, high-quality samples yields greater performance gains than indiscriminate data scaling.

### 2.2. Target-Aware Selection Mechanisms

To enable more granular control over data utility, researchers have adopted influence functions and gradient-based attribution methods to estimate the impact of individual samples on model performance (Koh & Liang, 2017; Wang et al., 2018; Kolossov et al., 2024). While classical influence functions provide a theoretical basis, their practical application is often limited by computational costs and robustness issues (Schioppa et al., 2023; Basu et al., 2021). Consequently, efficient approximations like Grad-Sim (Pruthi et al., 2020)

and TRAK (Park et al., 2023) have been developed to trace gradient similarities between training and reference examples. Building on these, LESS (Xia et al., 2024) introduces gradient projection and second-moment correction to enhance selection accuracy for instruction tuning, while other works focus on distribution alignment (Xie et al., 2023; Engstrom et al., 2024) or online selection (Wang et al., 2024). However, unlike LESS or GradSim, which rely on external gradient similarity metrics to infer relevance, our proposed TarGATE leverages the base model's inherent data understanding capabilities. By integrating the distributional characteristics of both the candidate and reference sets directly into the model's internal structure via learnable gates, TarGATE achieves a more intrinsic form of target awareness and data filtering, effectively distinguishing valuable signals from noise through forward-pass activation rather than post-hoc gradient analysis.

## 2.3. Gating Mechanisms

Gating mechanisms employ learnable switches to regulate information flow, a concept originating from the *forget gates* in LSTMs (Hochreiter & Schmidhuber, 1997; Gers et al., 2000) and later adapted for depth in Highway Networks (Srivastava et al., 2015). In modern LLMs, multiplicative gating variants like GLU (Dauphin et al., 2017) and SwiGLU (Shazeer, 2020) have become standard for efficient feature selection, while Mixture-of-Experts (MoE) (Shazeer et al., 2017) utilizes gates for conditional token routing. Unlike these architectural applications designed for representation learning or inference efficiency, TarGATE repurposes the gating mechanism as a data-centric probe, leveraging the gate's activation to explicitly quantify the intrinsic semantic utility of training samples.

## 3. Methodology

This section introduces **TarGATE**, an efficient data selection architecture designed to filter a large-scale candidate pool of instruction data, denoted as $D_{\text{candidate}}$, based on alignment with a small reference set $D_{\text{ref}}$ that reflects target capabilities. Following prior works (Xia et al., 2024; Jain et al., 2025), we ensure $D_{\text{ref}}$ is strictly disjoint from the held-out test set used for evaluation. To acquire *target-aware* capabilities, TarGATE is optimized using $D_{\text{ref}}$ and a small subset $D_{\text{warmup}} \subset D_{\text{candidate}}$ uniformly sampled to proxy the candidate distribution. By leveraging the standard language modeling loss $\mathcal{L}_{\text{LM}}$ on data instance $x$, TarGATE learns to quantify the utility of each sample, enabling us to rank $D_{\text{candidate}}$ based on this metric and select the top-$K$ subset for downstream fine-tuning.

## 3.1. TarGATE: A Lightweight Gating Architecture

To leverage the inherent capability of pretrained foundation models for data evaluation, we propose a mechanism that transitions from standard indiscriminate token processing to dynamic information regulation. We specifically target the feed-forward network (FFN), identified by recent studies as the hub for knowledge storage and semantic understanding (Yao et al., 2024; Dong et al., 2025; Xu & Chen, 2025; Geva et al., 2021), in contrast to the context-focused attention layers. By modulating the FFN's output intensity, we can explicitly filter noise while preserving contextual dependencies. To this end, we introduce a lightweight gating module named **TarGATE**, as shown in Figure 2. Inserted as a bypass structure before each FFN layer, this minimal gate, consisting of a single linear projection and a sigmoid activation, translates the model's implicit judgments into explicit retention signals with negligible computational overhead.

Formally, let $\mathbf{h}_{l-1}$ denote the hidden states input to layer $l$. The intermediate representation obtained after standard pre-norm multi-head attention (Xiong et al., 2020) processing is denoted as $\mathbf{h}_l^{\text{attn}}$:

$$\mathbf{h}_l^{\text{attn}} = \mathbf{h}_{l-1} + \text{Attn}(\text{RMSNorm}(\mathbf{h}_{l-1})). \quad (1)$$

We then compute a scaling value for each token to determine its **I**nformation **R**etention **R**atio (**IRR**). The gating mechanism takes $\mathbf{h}_l^{\text{attn}}$ as input and projects it using a learnable parameter vector $\mathbf{w}_l$. This is followed by a sigmoid function $\sigma(\cdot)$ to map the output to the range $(0, 1)$:

$$s_{l,t} = \mathbf{w}_l^\top \text{RMSNorm}(\mathbf{h}_{l,t}^{\text{attn}}), \qquad g_{l,t} = \sigma(s_{l,t}). \quad (2)$$

Finally, the IRR $g_{l,t}$ is utilized to attenuate the magnitude of the FFN update. The hidden state update for layer $l$ is thus modified as:

$$\mathbf{h}_l = \mathbf{h}_l^{\text{attn}} + \mathbf{g}_l \odot \text{FFN}(\text{RMSNorm}(\mathbf{h}_l^{\text{attn}})), \quad (3)$$

where $\mathbf{g}_l$ represents the gate values broadcast along the hidden dimension.

Within this gated model architecture, the gate effectively serves as an information filter during the data processing stage. An IRR close to 1 indicates that the token contains valid information worthy of FFN processing. Conversely, a ratio close to 0 implies that the token contributes little to the learning objective and should be suppressed. This approach transforms the foundation model into a dynamic data evaluator, capable of assessing data utility at a fine-grained token level. In the subsequent section, we will detail how to optimize these gate parameters to achieve target-aware data selection.

## 3.2. Optimizing Gates for Target Awareness

To align the data filtering capability of the gating module with the foundation model's data understanding capability,

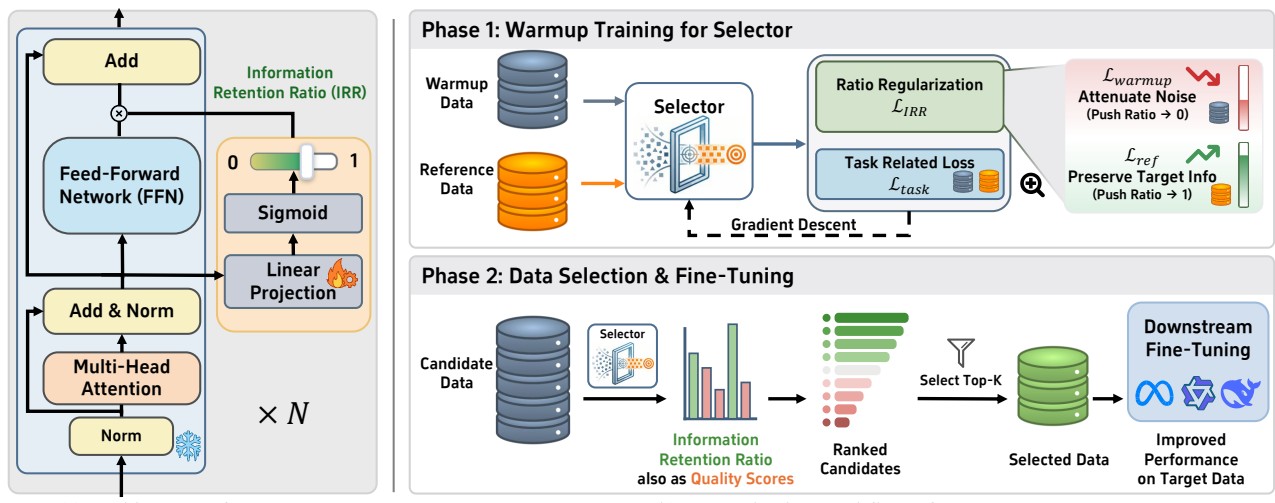

(a) Architecture of TarGATE    (b) Data selection workflow of TarGATE

*Figure 2.* **(a) Architecture of TarGATE.** A lightweight gating mechanism is integrated before each FFN layer. The gate employs a linear projection followed by a sigmoid activation to compute a token-level Information Retention Ratio (**IRR**), which dynamically modulates the information flow to filter out noise. **(b) Data selection workflow.** The workflow consists of two phases. In *Phase 1*, the selector is optimized via a dual-objective strategy while the backbone remains frozen: Ratio Regularization ($\mathcal{L}_{warmup}$) suppresses noise from general warmup data (pushing IRR $\rightarrow$ 0), while the Reference Loss ($\mathcal{L}_{ref}$) maximizes retention for task-specific reference data (pushing IRR $\rightarrow$ 1). In *Phase 2*, the massive candidate pool is evaluated via a single forward pass, and examples are ranked based on their quality scores (average IRR) to select the top-$K$ subset for downstream fine-tuning.

we propose a specialized optimization strategy. Our primary objective is to enable the gates to accurately perceive data distributions and efficiently filter out information that is unhelpful to the model's learning process. Utilizing the standard self-supervised learning task—specifically the language modeling loss as the fundamental training objective, we freeze all parameters of the backbone network, optimizing solely the gate parameters. During this warmup training phase, we randomly sample a small subset from the large-scale candidate pool $D_{candidate}$ to serve as the training dataset $D_{warmup}$, where the quality of this data is mixed, containing everything from high-quality instructions to meaningless noise.

To enable the gates to distinguish between these disparate data points, we apply a penalty to the IRR for data originating from $D_{warmup}$. We first compute the layer-wise average IRR $\bar{g}_l$ for a given input sequence $x$ by averaging the gate values across all $T$ tokens in layer $l$:

$$\bar{g}_l = \frac{1}{T} \sum_{t=1}^{T} g_{l,t}. \tag{4}$$

Subsequently, we introduce a regularization term $\mathcal{L}_{warmup}$ by aggregating these averages across all $L$ layers. This encourages the gates to filter out more information, thereby effectively suppressing the information flow:

$$\mathcal{L}_{warmup} = \sum_{l=1}^{L} \bar{g}_l. \tag{5}$$

This regularization imposes a sparsity constraint on the gates. Under the joint supervision of $\mathcal{L}_{LM}$ and $\mathcal{L}_{warmup}$, the model is compelled to attenuate redundant or non-informative tokens to satisfy the regularization constraint, while selectively opening the gates only for tokens that significantly contribute to reducing the prediction loss.

However, relying solely on $D_{warmup}$ to optimize the gating module allows it to filter only general low-quality noise, leaving it lacking in sensitivity toward specific target tasks. To bridge this gap and integrate target-aware capabilities into the gated model, we incorporate the small reference dataset $D_{ref}$ into the warmup training, which embodies the specific domain knowledge we aim to extract from the candidate pool. In contrast to $D_{warmup}$, we aim for the gates to actively recognize and accept the patterns found within $D_{ref}$. Therefore, we apply an opposing regularization term, $\mathcal{L}_{ref}$, specifically to the reference data to penalize low IRRs and encourage the gates to maintain information flow for these samples:

$$\mathcal{L}_{ref} = \sum_{l=1}^{L} (1 - \bar{g}_l). \tag{6}$$

By maximizing the IRR of the reference data, we explicitly align the gates' definition of *high quality* with the distributional characteristics of the target task. Finally, we formulate the combined quality regularization term $\mathcal{L}_{IRR}$ as a piecewise function depending on the data source:

$$\mathcal{L}_{IRR} = \begin{cases} \beta \mathcal{L}_{warmup} & \text{if } x \in D_{warmup} \\ (2-\beta) \mathcal{L}_{ref} & \text{if } x \in D_{ref} \end{cases} \tag{7}$$

where $\beta \in [0, 2]$ controls the selection preference.

These two objectives act as opposing forces on the gate mechanism. The warmup loss drives the gates to close in order to suppress general noise, whereas the reference loss pushes them to open to retain target-specific information. By balancing these terms through $\beta$, we regulate the strictness of the filter and compensate for the quantity imbalance arising from uniform batch sampling. A lower value of $\beta$ prioritizes the reference distribution and enforces a rigorous matching criterion suitable for specific domains. Conversely, a higher value relaxes this constraint and allows for a broader selection of diverse data. This flexibility enables TarGATE to adapt effectively to various task requirements.

The final training objective sums the language modeling loss with this quality-aware regularization, controlled by a hyperparameter $\lambda$:

$$\mathcal{L}_{\text{total}} = \mathcal{L}_{\text{LM}} + \lambda \mathcal{L}_{\text{IRR}}. \qquad (8)$$

Through this optimization process, TarGATE learns to assign low scores to noisy or irrelevant samples from the candidate distribution while assigning high scores to samples that share semantic similarities with the target domain.

### 3.3. Selecting High-Quality Data for Fine-Tuning

After employing the optimization strategies detailed in Section 3.2, TarGATE effectively captures the distributional characteristics of the target task and acquires the ability to discern and filter low-quality data. We then utilize this optimized gated model to evaluate the entire candidate pool $D_{\text{candidate}}$. Specifically, we calculate a quality score for each data instance by performing a single forward pass, defining the score as the global average of the IRR across all layers and tokens. This score quantitatively reflects the amount of information the model deems relevant to the target domain.

Formally, we define the example score $q(x)$ as the aggregated gate activation:

$$q(x) \triangleq \frac{1}{L} \sum_{l=1}^{L} \bar{g}_l(x), \qquad (9)$$

where $\bar{g}_l(x)$ is computed as in Equation 4. Subsequently, we rank all candidate samples in descending order based on their quality scores and retain the top-$K$ instances to construct the final selected dataset $D_{\text{sel}}$:

$$D_{\text{sel}} = \left\{ x \in D_{\text{candidate}} \mid x \in \text{TopK}(q, K) \right\}. \qquad (10)$$

Finally, this subset $D_{\text{sel}}$ is employed as the training set for the downstream fine-tuning of the target model. It is important to emphasize that our framework allows for architectural flexibility; the model used for downstream fine-tuning does not need to be identical to, or share weights with, the selection model.

## 4. Experiments

### 4.1. Experimental Setup

**Dataset & Models.** We evaluate TarGATE across two distinct scenarios. First, the *Synthetic Noisy Scenario* assesses robustness using a candidate pool of 32k examples mixed from MATH, GSM8K, CommonsenseQA, and synthetic random-word noise (Hendrycks et al., 2021; Cobbe et al., 2021; Talmor et al., 2019). We select the top 25% of data to fine-tune models for CommonsenseQA and GSM8K. Second, the *Real-World Massive Scenario* utilizes 200k instruction pairs from Tulu-3 (Lambert et al., 2024). Here, we select the top 5% to evaluate on a broader set of tasks: CommonsenseQA, GSM8K, and MBPP (Austin et al., 2021). Specific reference set construction and more dataset details are provided in Appendix A. To verify scalability, we employ Qwen3-0.6B and Qwen3-1.7B as selectors and transfer the curated data to larger target models for fine-tuning, including Qwen3-1.7B, Qwen3-14B, and Llama3-8B (Yang et al., 2025; Team, 2024).

**Baselines.** We compare against three categories of methods: (1) **Heuristic**: **Random Selection** and **BM25** (Trotman et al., 2014), which ranks samples based on lexical overlap with the reference. (2) **Embedding-based**: **RepSim** (Pezeshkpour et al., 2021), measuring cosine similarity of final-token hidden states, and **RDS+** (Ivison et al., 2025), which improves this by utilizing position-weighted mean pooling. (3) **Gradient-based**: **Grad-Sim** (Pruthi et al., 2020), calculating gradient similarity between training and reference data; **LESS** (Xia et al., 2024), an advanced influence estimation method using LoRA and AdamW optimizer states; and **ToV** (Jain et al., 2025), which selects samples exhibiting the highest loss reduction on validation tasks.

**Implementation Details.** We use the AdamW optimizer (Loshchilov & Hutter, 2019) for all experiments. During the warmup phase, TarGATE updates only the lightweight gating parameters. In contrast, for baselines requiring a warmup training, we uniformly apply LoRA (Hu et al., 2022) to all attention modules with a rank of $r = 128$ and $\alpha = 512$. For the subsequent downstream fine-tuning, we consistently employ the same LoRA configuration ($r = 128, \alpha = 512$) across all methods to ensure fair comparison. We set the regularization weight $\lambda = 1$ by default and tune the balance factor $\beta \in \{1.5, 0.5, 0.1, 0.05\}$ to adapt to different task types. We report the average performance and standard deviation across three random seeds. Each seed is fixed for the entire pipeline, including warmup-data sampling, random-selection baselines, gate optimization, and downstream fine-tuning.

*Table 1.* Main results on the **Real-World Massive Scenario**. We select top-5% data from the Tulu 3 dataset using proxy models (0.6B and 1.7B) to fine-tune three different target models (Qwen3-1.7B, Qwen3-14B, and Llama3-8B). "Full Data" denotes fine-tuning on the entire 200k candidate pool. Except for full data fine-tuning, the best results are **bolded** and the second are underlined.

| Method | Qwen3-1.7B-Base | | | Qwen3-14B-Base | | | Llama3-8B | | | Avg |
|---|---|---|---|---|---|---|---|---|---|---|
| | CQA | GSM8K | MBPP | CQA | GSM8K | MBPP | CQA | GSM8K | MBPP | |
| Full Data | 74.4±0.4 | 57.3±1.2 | 47.4±0.8 | 87.0±0.1 | 84.1±0.9 | 73.4±0.4 | 69.5±1.3 | 42.9±2.0 | 48.4±0.5 | 64.9 |
| Random Avg | 72.7±0.4 | 58.8±1.3 | 54.0±0.8 | 86.7±0.4 | 78.5±2.2 | 73.2±0.6 | 57.6±1.1 | 47.5±1.2 | 49.6±0.5 | 64.3 |
| BM25 | 72.8±0.3 | 57.7±0.8 | 55.2±0.8 | 87.5±0.4 | 82.7±0.7 | 72.2±0.1 | 53.8±0.5 | 51.0±0.8 | 50.2±0.1 | 64.8 |
| | | | | *Data Selector: Qwen3-0.6B-Base* | | | | | | |
| GRADSIM | 72.4±0.5 | 59.6±1.4 | 54.8±0.4 | 86.2±0.3 | 81.8±1.6 | 73.2±0.8 | 63.1±0.9 | 56.3±1.4 | 48.2±0.3 | 66.2 |
| LESS | 73.0±0.3 | 58.6±0.4 | 54.8±1.9 | 86.4±0.6 | 80.5±2.2 | 72.6±0.4 | 62.1±1.2 | 54.2±0.5 | 48.8±0.1 | 65.7 |
| REPSIM | 72.7±0.4 | 62.6±1.6 | 52.2±0.5 | 86.4±0.2 | 79.6±1.4 | 73.4±0.7 | 60.4±0.6 | 56.2±1.2 | 48.6±0.3 | 65.8 |
| TOV | 73.2±0.5 | 58.1±0.9 | 51.0±0.8 | **87.3±0.4** | 82.9±0.9 | 73.2±0.3 | 64.4±1.4 | 52.2±2.3 | 49.6±0.2 | 65.7 |
| RDS+ | 73.0±0.3 | 61.9±1.3 | 55.2±1.4 | 86.3±0.3 | 80.6±1.8 | 73.0±0.6 | 62.9±1.2 | 57.9±0.9 | 49.2±0.1 | 66.6 |
| TarGATE | 73.7±0.4 | **62.1±0.7** | **57.2±0.6** | **87.8±0.4** | **84.4±1.2** | **73.8±0.3** | **65.8±0.8** | **58.2±0.7** | **51.0±0.3** | **68.2** |
| | | | | *Data Selector: Qwen3-1.7B-Base* | | | | | | |
| GRADSIM | 72.9±0.3 | 59.2±1.2 | 55.0±0.2 | 85.2±0.3 | 78.0±1.9 | 73.0±0.6 | 59.3±1.3 | 45.7±1.3 | 49.0±0.2 | 64.1 |
| LESS | 72.6±0.4 | 55.8±0.5 | 53.8±1.6 | 85.7±0.2 | 80.9±1.2 | 72.6±0.5 | 59.5±1.7 | 49.4±1.0 | 50.0±0.1 | 64.5 |
| REPSIM | 73.1±0.2 | 59.3±1.6 | 54.0±1.2 | 86.9±0.3 | 79.9±0.5 | 73.0±0.2 | 56.7±1.8 | 45.9±1.4 | 48.8±0.2 | 64.2 |
| TOV | 72.5±0.2 | 57.7±0.6 | 50.8±0.6 | 86.8±0.2 | 80.6±0.8 | 72.4±0.9 | 54.0±1.5 | 43.5±2.2 | 49.4±0.4 | 63.1 |
| RDS+ | 72.6±0.4 | 60.7±1.2 | 54.6±1.7 | 86.3±0.5 | 81.5±1.2 | 73.0±0.3 | 61.5±1.2 | 53.6±0.6 | 50.0±0.3 | 65.9 |
| TarGATE | **74.1±0.3** | 61.1±0.9 | 56.4±0.4 | 87.2±0.6 | **85.5±1.3** | 73.6±0.5 | 63.2±1.4 | 57.4±0.7 | 50.0±0.2 | 67.6 |

*Table 2.* Performance comparison on the **Synthetic Noisy Scenario**. The candidate pool consists of 32k samples mixed with synthetic noise. We report the test accuracy (%) on CommonsenseQA (CQA) and GSM8K. The best results are **bolded** and the second best are underlined.

| Method | Qwen3-1.7B-Base | | Llama3-8B | | Avg |
|---|---|---|---|---|---|
| | CQA | GSM8K | CQA | GSM8K | |
| Random | 74.2±0.28 | 54.7±0.87 | 71.6±0.45 | 42.1±0.70 | 60.6 |
| BM25 | 74.1±0.48 | 54.8±0.66 | 72.9±0.90 | 37.9±2.20 | 59.9 |
| | *Data Selection via Proxy: Qwen3-0.6B-Base* | | | | |
| GRADSIM | 72.8±0.1 | 55.4±0.2 | 71.1±0.1 | 36.1±1.21 | 58.8 |
| LESS | 72.4±0.7 | 54.1±1.0 | 70.8±0.2 | 42.6±0.4 | 60.0 |
| REPSIM | 72.4±0.1 | 52.0±0.4 | 70.5±0.1 | 40.5±1.4 | 58.8 |
| TOV | 78.9±0.6 | 53.9±1.0 | 81.4±0.1 | 39.0±1.0 | 63.3 |
| RDS+ | 71.5±0.9 | 55.9±0.7 | 71.4±0.8 | 38.2±0.5 | 59.2 |
| TarGATE | **79.5±0.5** | **57.9±0.7** | **81.8±0.2** | 47.0±0.5 | **66.5** |
| | *Data Selection via Proxy: Qwen3-1.7B-Base* | | | | |
| GRADSIM | 75.3±0.6 | 54.4±0.3 | 72.2±0.2 | 43.0±0.3 | 61.2 |
| LESS | 75.3±0.5 | 54.5±1.2 | 71.4±0.2 | 40.0±0.6 | 60.3 |
| REPSIM | 75.9±1.1 | 53.1±0.9 | 70.8±1.2 | 40.4±1.4 | 60.1 |
| TOV | 78.6±0.7 | 51.8±0.4 | 81.2±0.1 | 39.3±1.0 | 62.7 |
| RDS+ | 71.9±0.4 | 54.6±1.1 | 72.4±0.1 | 35.8±0.9 | 58.7 |
| TarGATE | 79.0±0.6 | 56.7±0.3 | 81.7±0.1 | **48.6±0.8** | **66.5** |

## 4.2. Overall Performance

Tables 1 and 2 summarize the results. In the Synthetic Noisy Scenario (Table 2), TarGATE demonstrates significant robustness, achieving the highest average score of 66.5% with both Qwen3-0.6B and 1.7B selectors. It substantially outperforms Random (60.6%) and BM25 (59.9%), indicating effective noise filtering. Furthermore, compared to gradient-based methods like LESS and GradSim, TarGATE maintains a clear lead on both CommonsenseQA and GSM8K, suggesting that directly optimizing information flow provides a more reliable metric than indirect influence estimation in

noisy environments.

Moving to the Real-World Massive Scenario shown in Table 1, TarGATE continues to exhibit superior performance. The data chosen by TarGATE consistently yields better downstream fine-tuning results than using the full dataset score of 64.9% or the random sampling score of 64.3% which effectively validates the premise that training on a curated subset can surpass training on the entire corpus. Notably, under the Qwen3-0.6B selector setting, TarGATE achieves the highest average score of 68.2% and comprehensively surpasses strong baselines such as TOV and RDS+ across diverse domains including CQA, GSM8K, and MBPP. This indicates that TarGATE can precisely locate high-quality samples most valuable to the target task from massive heterogeneous data sources.

The results also highlight the superior stability of TarGATE across different task types compared to influence-based and representation-based methods. While methods like LESS perform competitively on reasoning tasks like GSM8K, they often struggle to generalize to code generation tasks like MBPP where TarGATE maintains a distinct advantage as evidenced by the 51.0% score compared to 48.8% for LESS with the 0.6B selector. We attribute this to the fact that our gating mechanism is optimized via the target distribution to avoid the potential biases found in gradient approximations or static embedding similarities. Consequently, TarGATE ensures consistent superiority across diverse tasks ranging from common sense reasoning to code generation.

Finally, we observe that the smaller 0.6B selector often yields slightly better downstream performance compared to the 1.7B selector with an average score of 68.2% versus

67.6%. This phenomenon has been reported in prior studies such as LESS and DATE-LM which attribute it to stronger models being more susceptible to surface-level stylistic features whereas weaker models tend to select data with higher transferability. However, TarGATE demonstrates greater robustness to the selector scale. For instance, when using the 1.7B model to select data for Llama3-8B on GSM8K, TarGATE maintains a high score of 57.4% whereas other baselines like GradSim exhibit a significant performance drop down to 45.7%. This further indicates that the target-aware capability acquired by TarGATE is driven by intrinsic semantic relevance rather than model-specific biases.

### 4.3. Distributions of Selected Data

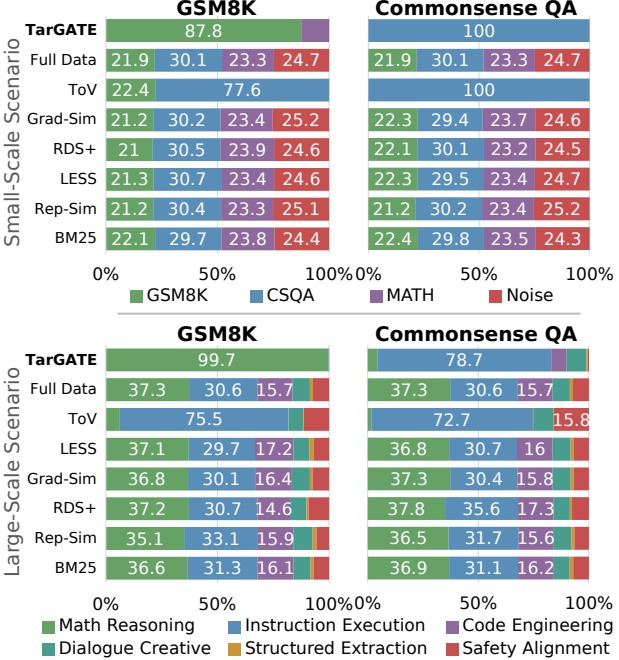

Figure 3 illustrates the categorical distribution of data selected by different methods using the Qwen3-0.6B model as the selector, where the data categories for the large-scale scenario were labeled by GLM-4.7 (Z.AI, 2025). An evident trend across both scale settings is that, with the exception of TarGATE and ToV, the subsets selected by most baseline methods closely mirror the composition of the Full Data, failing to exhibit distinct distribution shifts in response to changes in the target task. While ToV proves effective at filtering out synthetic noise in the synthetic scenario, it does not demonstrate a proactive preference for domain-specific data, as evidenced by the lack of increased mathematical

samples for the GSM8K task. In contrast, TarGATE exhibits strong adaptability to the target requirements; specifically, by setting $\beta = 0.1$ for the reasoning-intensive GSM8K task and $\beta = 1.5$ for Commonsense QA, TarGATE successfully aligns its selection focus with the target capabilities, retrieving a significantly higher proportion of math-reasoning data for GSM8K compared to all other methods.

### 4.4. Distributions of Data Scores

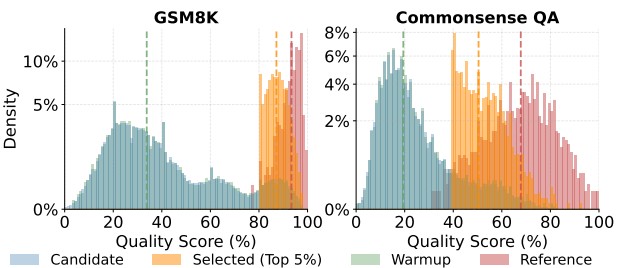

*Figure 4.* Distribution of quality scores for GSM8K and Commonsense QA. Dashed vertical lines indicate the mean score for each group. The distributions show that the randomly sampled Warmup set effectively represents the Candidate pool, while the Selected data (Top 5%) shifts significantly towards the high-scoring Reference distribution.

To intuitively understand how TarGATE evaluates data importance, Figure 4 visualizes the distributions of quality scores computed by the Qwen3-0.6B selector. These results correspond to the optimal hyperparameters identified in Table 1, specifically regarding the balance factor $\beta$. A distinct observation is the near-perfect overlap between the score distributions of the massive Candidate pool and the small, randomly sampled Warmup set. This confirms that our lightweight warmup strategy, which relies on a random subset, effectively captures the global distribution characteristics of the large-scale dataset. Furthermore, the dashed vertical lines, representing the mean scores of each subset, reveal a clear quality hierarchy: the Candidate set exhibits the lowest average quality, while the Reference set consistently occupies the highest score range. Notably, the mean score of the data selected by TarGATE (Selected) resides between these two extremes but is significantly shifted towards the Reference mean. This demonstrates that TarGATE, through optimization, successfully learns to treat reference-aligned patterns as high-quality signals, thereby effectively identifying and retrieving data relevant to the target task from a noisy candidate pool.

The varying shapes of the distributions further elucidate the model's preference for different $\beta$ parameters across tasks. For GSM8K, which demands rigorous mathematical reasoning, a smaller $\beta = 0.1$ is preferred. This stricter constraint forces the model to prioritize data highly similar to the reference, resulting in a selected distribution that is densely

concentrated in the high-score region. Conversely, for Commonsense QA, which benefits from broader general knowledge, a larger $\beta = 1.5$ is employed. This relaxed constraint reduces the penalty on non-reference patterns, leading to a more spread-out distribution for the selected data.

### 4.5. Ablation on Hyperparameters

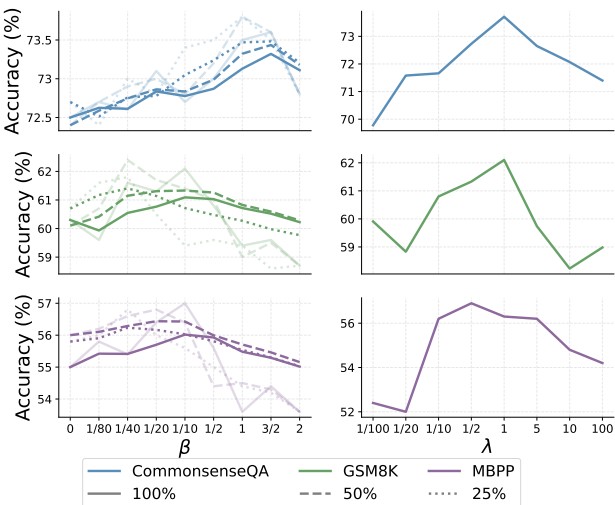

*Figure 5.* Ablation study on hyperparameters. The left column shows the impact of $\beta$ under different reference set sizes (100%, 50%, 25%), with smoothed trends plotted over transparent raw values. The right column illustrates the robustness of $\lambda$ across tasks. Specialized tasks and smaller reference sets generally favor lower $\beta$, while $\lambda = 1$ proves effective across diverse scenarios.

We conduct ablation studies using Qwen3-0.6B as the selector and Qwen3-1.7B as the target model to evaluate the impact of hyperparameters $\beta$ and $\lambda$. We first examine the extreme settings of $\beta = 0$ and $\beta = 2$, which correspond to optimizing solely through $\mathcal{L}_{ref}$ or $\mathcal{L}_{warmup}$, respectively. The significant performance drops observed at these boundaries in Figure 5 demonstrate that neither objective is sufficient in isolation. Crucially, TarGATE relies on the joint optimization of these losses to effectively contrast the target distribution captured by $\mathcal{L}_{ref}$ against the general candidate distribution represented by $\mathcal{L}_{warmup}$. This mechanism enables the gates to perceive distribution shifts and acquire true target-aware discrimination capabilities.

Regarding the sensitivity to data availability and task types, we vary the reference set size at 100%, 50%, and 25% levels. The results reveal a distinct task-dependent preference for $\beta$. CommonsenseQA achieves optimal performance with larger $\beta$ values, as general knowledge tasks benefit from a relaxed filter that retains diverse information. Conversely, GSM8K and MBPP favor smaller $\beta$ values, as reasoning and coding tasks demand strict adherence to specific logic patterns found in the reference data. Furthermore, reducing the reference set size generally shifts the optimal $\beta$ toward

smaller values to compensate for the weaker target signal by enforcing tighter selection constraints. Consequently, we recommend tuning $\beta$ based on task specificity and data abundance.

The analysis of the regularization weight $\lambda$ indicates that optimal performance is consistently achieved near 1.0 or 0.5 across tasks. Performance drops significantly when $\lambda$ becomes extremely large or small, as excessive regularization disrupts the foundation model's language understanding while insufficient regularization fails to impose the necessary sparsity for effective filtering. These findings suggest that $\lambda$ is best tuned within the range of $[0.1, 10]$ to balance these objectives. Our default setting of $\lambda = 1$ demonstrates robust performance without requiring extensive task-specific adjustment.

### 4.6. Complexity Analysis

We quantitatively estimate the computational overhead using Qwen3-1.7B as the selector following the DATE-LM protocol (Jiao et al., 2025). The analysis aggregates FLOPs incurred during both the *Warmup* phase and the *Selection* phase over 200k samples. Detailed calculation protocols are provided in Appendix D.

*Table 3.* Complexity comparison using Qwen3-1.7B. *Params* denotes trainable parameters in warmup. Costs are in PetaFLOPs (PF) for 200k samples.

| Method | #Params | Warmup | Selection | Total |
|---|---|---|---|---|
| Random | - | - | $\approx 0.0$ | $\approx 0.0$ |
| BM25 | - | - | $\approx 0.0$ | $\approx 0.0$ |
| Rep-Sim | - | - | 352.8 | 352.8 |
| RDS+ | - | - | 352.8 | 352.8 |
| Grad-Sim | 25.7 M | 70.5 | 705.5 | 776.0 |
| LESS | 25.7 M | 70.5 | 705.5 | 776.0 |
| ToV | 25.7 M | 72.6 | 1409.0 | 1481.6 |
| **TarGATE** | **0.05M** | 72.6 | 352.3 | **424.9** |

As shown in Table 3, heuristic baselines incur negligible costs, while gradient-based methods (Grad-Sim, LESS) and ToV remain computationally intensive ($\sim$776–1482 PF) due to the necessity of backward passes or multiple inference rounds. In contrast, TarGATE achieves superior efficiency (424.9 PF) by performing selection via a single forward pass. Even accounting for the reduced backward-pass overhead of LoRA-based implementations, TarGATE yields an efficiency gain of approximately $1.8\times$ over gradient-based approaches.

## 5. Limitations

TarGATE is designed as a target-aware selector and therefore depends on the quality and coverage of the reference set; noisy or poorly matched references can weaken the learned selection preference. Although the IRR score provides a

useful internal signal for ranking examples, our analysis of token-level gating behavior remains preliminary and does not yet fully explain which linguistic or reasoning patterns are suppressed in all settings. Finally, our experiments focus on targeted instruction tuning rather than general-purpose data filtering, so extending the method to select data that benefits a broad suite of downstream tasks remains an important direction for future work.

## 6. Conclusion

In this paper, we address the challenge of efficient data selection for targeted instruction tuning. We propose **TarGATE**, a novel framework that leverages the data understanding capabilities of foundation models to discern high-quality training examples. By introducing a lightweight gating mechanism and employing a dual-objective optimization strategy that contrasts general noise with target-specific references, we enable the model to autonomously quantify data relevance through internal information flow. Extensive experiments across noisy and real-world scenarios demonstrate that TarGATE consistently outperforms SOTA baselines in downstream performance, while offering significant advantages in efficiency and cross-model transferability.

## Acknowledgments

This work was supported by the National Key Research and Development Program of China under grant 2024YFC3307900; the National Natural Science Foundation of China under grants 62302184, 62376103, 62436003, and 62206102; the Major Science and Technology Project of Hubei Province under grant 2024BAA008; the Hubei Science and Technology Talent Service Project under grant 2024DJC078; Ant Group through the CCF-Ant Research Fund; and the Fundamental Research Funds for the Central Universities under grant YCJJ20252319. The computations were performed on the HPC Platform of Huazhong University of Science and Technology.

## Impact Statement

This paper introduces a framework for efficient data selection, aimed at improving the training efficiency of Large Language Models. By enabling models to reach target performance with significantly less data, our method contributes to reducing the carbon footprint associated with training large-scale AI systems. While we do not foresee immediate negative societal consequences, we emphasize that the ethical alignment of the selected data remains dependent on the integrity of the provided reference examples.

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

# A. Experimental Setup Details

In this section, we provide a detailed breakdown of the dataset construction and the hyperparameters used for both the TarGATE warmup phase and the downstream fine-tuning phase.

## A.1. Dataset Construction

**Synthetic Noisy Candidate Pool.** To evaluate the robustness of data selection methods against explicit noise and mixed distributions, we constructed a synthetic candidate pool containing 32,341 samples. We uniformly sample 8k data points from the candidate pool as the $D_{\text{warmup}}$. This pool is a composite of four distinct data sources:

- **GSM8K (Math Reasoning)** (Cobbe et al., 2021): We utilize the training split of the GSM8K dataset. To strictly avoid data leakage with the reference set, we select only the first 95% of the training examples, contributing 7,100 samples to the candidate pool.

- **MATH (Math Reasoning)** (Hendrycks et al., 2021): We incorporate 7,500 samples randomly sampled from the MATH training set to increase the diversity of mathematical reasoning data.

- **CommonsenseQA (General Reasoning)** (Talmor et al., 2019): We include 9,741 samples from the training split of CommonsenseQA, representing general knowledge and logical reasoning capabilities.

- **Synthetic Noise**: To simulate low-quality or irrelevant data often found in large-scale web corpora, we generated 8,000 samples using the Python `faker` library. These samples consist of random sequences of words and carry no semantic meaning.

**Real-World Massive Candidate Pool.** Following the experimental protocols of DATE-LM (Jiao et al., 2025), we adopt the **Tulu-3** dataset (Lambert et al., 2024) as our real-world instruction tuning source. From the full dataset, we randomly sample 200,000 examples to serve as the candidate pool for selection. We uniformly sample 10k data points from the candidate pool as the $D_{\text{warmup}}$.

**Reference Sets and Target Tasks.** To guide the target-aware selection, we construct distinct reference sets ($D_{\text{ref}}$) for three different downstream capabilities: mathematical reasoning, code generation, and general common sense reasoning.

- **GSM8K (Math)**: We utilize the last 5% of the GSM8K training set (distinct from the portion used in the synthetic pool) as the reference set, comprising 373 samples. This ensures that the reference distribution captures the specific style of grade-school math problems.

- **MBPP (Code)** (Austin et al., 2021): We employ the full training set of the MBPP dataset, consisting of 374 samples, to represent code generation capabilities.

- **CommonsenseQA (General)**: We use the validation set of CommonsenseQA, containing 1,221 samples, as the reference for general reasoning tasks.

These three tasks were selected to comprehensively cover distinct model capabilities: multi-step mathematical reasoning, structured code engineering, and broad-domain conversational reasoning. All downstream evaluations are conducted using the Language Model Evaluation Harness (Gao et al., 2024).

## A.2. Training and Evaluation Details

We aligned our experimental configuration with the setup described in DATE-LM to ensure fair comparison. The training process consists of two stages: the warmup of the selector (TarGATE or baselines) and the downstream fine-tuning of the target model.

**Warmup Phase.** In the warmup phase, we train for 2 epochs. We employ an AdamW optimizer (Loshchilov & Hutter, 2019) with a linear learning rate scheduler. The global batch size is set to 128, and the maximum sequence length is set to 1024 tokens.

- **Baselines:** For baseline methods requiring a warmup training, we apply LoRA (Hu et al., 2022) fine-tuning to the attention modules with a rank of $r = 128$, $\alpha = 512$, and a peak learning rate of 2e-5.

- **TarGATE:** For our proposed method, we freeze the backbone model and optimize *only* the gate parameters. We use a higher peak learning rate of 3e-4 to ensure effective convergence of the gates. The reference and warmup samples are combined into a unified training pool. We employ a standard random shuffling strategy to construct batches, without enforcing specific mixing ratios.

- **LESS (Xia et al., 2024):** We adopt the simplified evaluation protocol established in DATE-LM (Jiao et al., 2025). We compute similarity using raw, unprojected gradients rather than performing low-rank random projections, as high-dimensional projections incur additional computational costs without performance gains in this setting.

- **ToV (Jain et al., 2025):** The ToV algorithm relies on a hyperparameter $L$, which designates the number of training loops performed to iteratively track the loss differential of candidate samples before and after validation fine-tuning. The final importance score is calculated by averaging these differentials across the $L$ epochs. To prevent the data selection phase from incurring excessive computational costs and to maintain a fair comparison with other baselines, we set $L = 2$.

**Downstream Fine-tuning Phase.** For the final supervised fine-tuning (SFT) on the selected data, we consistently apply the following hyperparameters across all methods and models:

- **Optimization:** We train for 1 epoch using the AdamW optimizer (Loshchilov & Hutter, 2019) and a linear learning rate scheduler with a peak learning rate of 5e-5.

- **Hyperparameters:** The global batch size is maintained at 128, with a maximum sequence length of 1024 tokens.

- **LoRA Configuration:** To efficiently fine-tune the target models, we apply LoRA (Hu et al., 2022) to the attention modules. We set the LoRA rank to $r = 128$ and $\alpha = 512$ to provide sufficient capacity for learning from the curated data.

## B. Additional Experimental Results

### B.1. Impact of Training Configurations

In this section, we present additional experimental results regarding the number of fine-tuning epochs and the scale of the selected data. All experiments reported here are conducted using Qwen3-0.6B as the selector and Qwen3-1.7B as the target model on the MBPP task under the Real-World Massive Scenario.

**Impact of Fine-tuning Epochs.** We evaluated the downstream performance across different fine-tuning epochs, ranging from 1 to 4. As shown in Table 4, we observe a general performance decline across most methods as the number of epochs increases. We attribute this simply to overfitting. Since the selected datasets are relatively small (typically 5% of the candidate pool), training for multiple epochs causes the model to overfit to the limited data samples, which harms its generalization capability. Therefore, fine-tuning for 1 epoch is sufficient to reflect the quality of the selected data while maintaining computational efficiency.

**Impact of Selected Data Scale.** We also investigated the effect of varying the selected data size from 10k (5%) to 40k (20%). The results in Table 5 show that increasing the volume of selected data generally leads to a decrease in model performance. This indicates that the top-ranked data (the top 5%) contains the most relevant information for the target task. Extending the selection range inevitably includes candidates with lower relevance scores, which acts as noise and degrades the overall fine-tuning effect. Consequently, we strictly adhere to the 5% selection ratio in our main experiments.

### B.2. Impact of Reference Data in Warmup

To determine whether the performance advantage of TarGATE stems primarily from the exposure to reference data during the warmup phase, we conducted an additional ablation study where the baseline methods, LESS and Grad-Sim, were also optimized using a mixture of the reference and warmup sets. We performed this experiment using the Qwen3-0.6B model

*Table 4.* Ablation study on the number of fine-tuning epochs. We report the pass@1 accuracy (%) on MBPP. Performance drops with more epochs due to overfitting on the small-scale selected data.

| Method | Epochs | | | |
|---|---|---|---|---|
| | **1** | **2** | **3** | **4** |
| Random | 54.6 | 46.2 | 43.8 | 43.6 |
| BM25 | 55.2 | 52.2 | 52.0 | 52.2 |
| Grad-Sim | 54.8 | 54.2 | 53.4 | 47.8 |
| LESS | 54.8 | 53.0 | 53.4 | 46.8 |
| Rep-Sim | 52.2 | 51.0 | 49.6 | 45.8 |
| ToV | 51.0 | 48.8 | 48.2 | 43.4 |
| RDS+ | 55.2 | 52.0 | 52.0 | 50.8 |
| **TarGATE** | **57.2** | **55.0** | **55.4** | **54.4** |

*Table 5.* Ablation study on the size of the selected dataset. We report the pass@1 accuracy (%) on MBPP. Increasing the data scale generally reduces performance as it introduces lower-quality samples.

| Method | Selected Data Size | | | |
|---|---|---|---|---|
| | **10k (5%)** | **20k (10%)** | **30k (15%)** | **40k (20%)** |
| Random | 54.6 | 54.6 | 54.4 | 51.6 |
| BM25 | 55.2 | 53.2 | 54.0 | 53.2 |
| Grad-Sim | 54.8 | 53.6 | 54.2 | 52.8 |
| LESS | 54.8 | 53.0 | 53.2 | 52.8 |
| Rep-Sim | 52.2 | 52.4 | 51.6 | 51.4 |
| ToV | 51.0 | 44.8 | 48.6 | 47.6 |
| RDS+ | 55.2 | 52.8 | 51.0 | 46.4 |
| **TarGATE** | **57.2** | **55.0** | **55.8** | **55.2** |

as the selector and Qwen3-1.7B as the target model within the Real-World Massive Scenario. As presented in Table 6, providing LESS and Grad-Sim with early access to the reference set during the warmup stage did not yield significant performance improvements compared to their standard configurations where reference data is only introduced during the selection phase. Consequently, to ensure a standard and fair comparison, we adhered to the original settings for these baselines in our main experiments, utilizing the reference set solely for gradient or influence estimation after the warmup is complete.

*Table 6.* Performance comparison when LESS and Grad-Sim are trained with the reference set included in the warmup phase (w/ Ref Warmup). Experiments are conducted with Qwen3-0.6B as the selector and Qwen3-1.7B as the target model.

| Method | COMMON | GSM8K | MBPP |
|---|---|---|---|
| LESS | 72.8 | 59.2 | 54.8 |
| Grad-Sim | 72.9 | 59.8 | 54.4 |
| TarGATE | 73.7 | 62.1 | 57.2 |

### B.3. More Distributions of Data Scores

We expand our analysis of data quality evaluation by presenting the score distributions for three distinct downstream tasks and two different selector scales in Figure 6. This visualization encompasses the tasks of GSM8K, Commonsense QA, and MBPP, utilizing both Qwen3-0.6B and Qwen3-1.7B as the backbone models for data selection. A consistent pattern emerges across all experimental settings where the randomly sampled Warmup set exhibits a distribution that is nearly indistinguishable from the massive Candidate pool. This overlap empirically validates our methodological assumption that a small random subset suffices to capture the global characteristics of the noise distribution. In sharp contrast, the Reference data consistently occupies the upper bound of the quality spectrum and serves as a distinct anchor for high-utility information. The Selected data distribution successfully bridges this gap by shifting significantly away from the low-quality Candidate region and aligning closely with the high-scoring Reference domain.

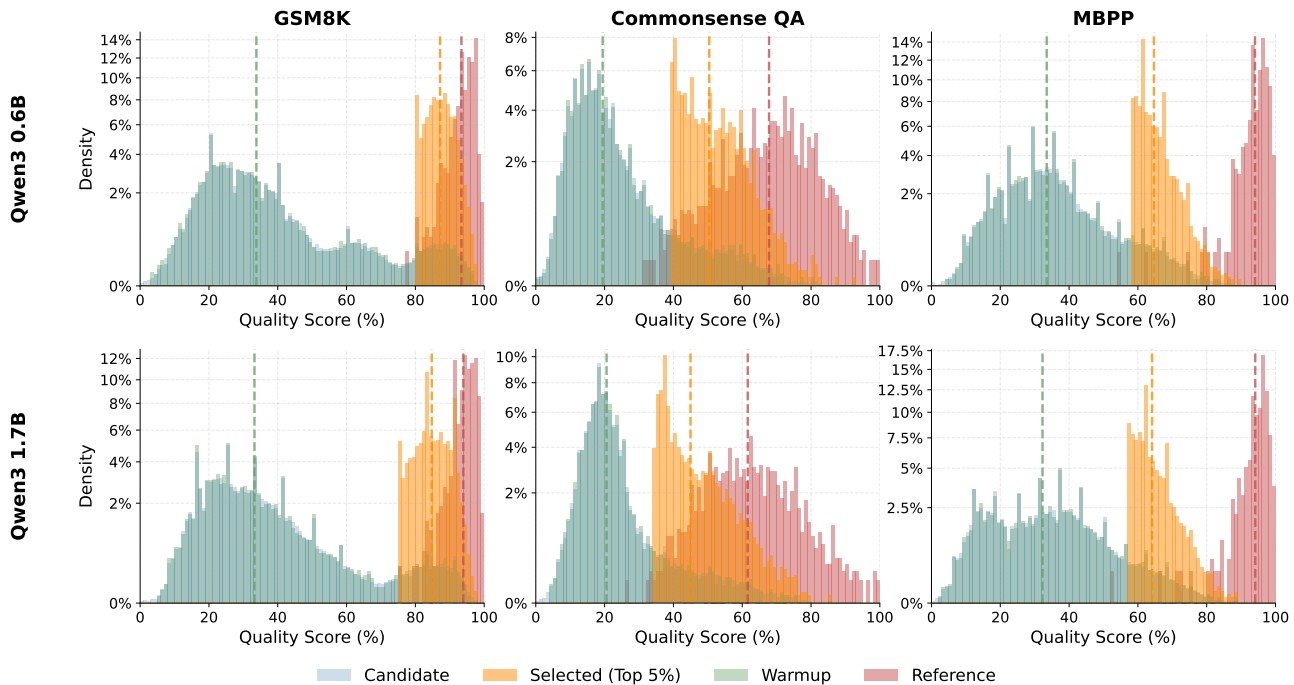

*Figure 6.* Visualization of quality score distributions across different selector scales and target tasks. The top and bottom rows display results for Qwen3-0.6B and Qwen3-1.7B respectively. Vertical dashed lines represent the mean score for each data subset. The distributions demonstrate that TarGATE effectively identifies high-quality data by shifting the selection statistics from the random Warmup distribution toward the target-aligned Reference distribution.

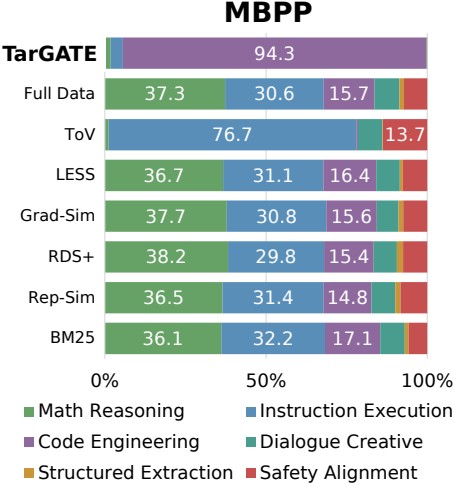

*Figure 7.* The distribution of data categories selected by different methods for the MBPP task using Qwen3-0.6B as the selector. While baselines mirror the full data and ToV drifts towards instruction following, TarGATE achieves a dominant concentration of Code Engineering data.

## B.4. Distributions of Selected Data on MBPP

Figure 7 displays the categorical distribution of data selected for the MBPP task using the Qwen3-0.6B selector. Similar to the reasoning tasks, most baselines including LESS and Grad-Sim exhibit a distribution nearly identical to the Full Data and fail to increase the proportion of Code Engineering samples beyond the baseline level of 15.7%. While ToV selects a distinct subset, it demonstrates a clear misalignment with the coding objective by predominantly selecting Instruction Execution data rather than programming tasks. In contrast, TarGATE exhibits precise target awareness by filtering out diverse noise to

achieve a 94.3% concentration of Code Engineering data which validates its robust capability to identify domain-specific semantics within a heterogeneous candidate pool.

## C. Baselines Details

In this section, we provide detailed descriptions of the baseline methods employed in our evaluation. These methods span three primary categories: heuristic approaches, representation-based methods, and gradient-based strategies.

**BM25 (Trotman et al., 2014).** BM25 (Best Matching 25) is a classical probabilistic information retrieval algorithm. In the context of data selection, it treats the reference dataset as a query and ranks samples from the candidate pool based on lexical overlap. It calculates a relevance score that accounts for term frequency and inverse document frequency, effectively selecting training examples that share significant textual similarities with the target domain. Due to its reliance on surface-level statistics rather than neural embeddings, BM25 is computationally efficient but lacks semantic understanding.

**Rep-Sim (Pezeshkpour et al., 2021).** Representation Similarity (Rep-Sim) is a non-gradient embedding-based method. It measures the semantic proximity between candidate samples and the reference set by utilizing the model's intermediate representations. Specifically, Rep-Sim computes the cosine similarity between the last-layer hidden state of the final token for both the training and reference examples. This approach assumes that the final token's embedding aggregates the semantic information of the entire sequence.

**RDS+ (Ivison et al., 2025).** RDS+ is an enhanced variant of representation-based selection. While Rep-Sim relies solely on the final token, RDS+ seeks to capture a richer semantic profile of the data. It computes a position-weighted mean pool of the last hidden layer states across all tokens in the sequence. By aggregating information from the entire context window, RDS+ provides a more robust embedding for calculating similarity against the reference distribution.

**Grad-Sim (Pruthi et al., 2020).** Grad-Sim estimates the influence of a training example by analyzing first-order gradient information. It computes the gradient of the loss with respect to the model parameters for both the candidate data and the reference data. The influence score is then defined as the cosine similarity between these two gradient vectors. Intuitively, this method selects data that updates the model parameters in a direction similar to the updates required by the reference task.

**LESS (Xia et al., 2024).** LESS (Low-rank Gradient Error selection via Second-order Approximation) builds upon the principles of gradient similarity but introduces optimizations for accuracy and efficiency. To better approximate the change in training loss, LESS incorporates second-order information by utilizing the Adam optimizer states (gradient moments) to re-weight the gradients. Furthermore, to handle the high dimensionality of LLM parameters, it projects gradients into a lower-dimensional space using random projections and leverages LoRA for efficient computation. This allows for a more faithful estimation of data influence compared to standard first-order methods.

**ToV (Jain et al., 2025).** Training on Validation (ToV) identifies high-quality data by measuring the impact of fine-tuning on the target distribution. Following the official implementation (Method A), the algorithm first trains a base model on a randomly sampled subset of the candidate pool (referred to as the base subset or warmup set $D_{\text{warmup}}$) to establish a training trajectory. Subsequently, it fine-tunes this model on the reference set ($D_{\text{ref}}$). The importance score is derived from the loss discrepancy of candidate samples evaluated between the base model and the reference-tuned model. Accordingly, the computational cost includes gradients computations for both the warmup phase on $D_{\text{warmup}}$ and the adaptation phase on $D_{\text{ref}}$. We set the number of training loops $L = 2$.

## D. Computational Cost Estimation Details

To fairly evaluate the efficiency of TarGATE compared to other data selection baselines, we provide a detailed theoretical breakdown of the Floating Point Operations (FLOPs). We follow (Chen et al., 2025; Jiao et al., 2025) to derive the FLOPs for forward and backward passes.

**FLOPs Derivation.** Consider a single linear layer with input dimension $n$ and output dimension $m$. For standard training, the forward pass requires $2bmn$ FLOPs, where $b$ is the batch size. In contrast, Parameter-Efficient Fine-Tuning (PEFT)

methods used in our experiments introduce a negligible amount of additional parameters. For LoRA baselines with rank $r$, the additional forward cost scales with $r$. For TarGATE, the gates project hidden states to a single scalar, which implies an output dimension of 1. Since both $r \ll m, n$ and $1 \ll m, n$, the additional forward pass cost for both methods is negligible compared to the base model projection. Thus, the forward pass cost remains approximately $2bmn$.

For the backward pass, standard training incurs a cost of $4bmn$ ($2bmn$ for input gradient calculation $\nabla_x \mathcal{L}$ and $2bmn$ for weight gradient calculation $\nabla_W \mathcal{L}$). However, for the efficient methods used here, we only need to compute gradients for a sparse set of parameters. The cost for input gradient calculation remains $2bmn$ (to propagate error), but the weight gradient calculation is drastically reduced. For LoRA, this term is proportional to $r$; for TarGATE, it is proportional to 1. In both cases, the weight gradient overhead is mathematically negligible. Consequently, we approximate the backward pass cost to be identical to the forward pass. Based on this derivation, we estimate the cost of an inference step as $2N$ FLOPs per token, and a training step (forward + backward) as $4N$ FLOPs per token, where $N$ is the model parameter count.

**Dataset and Model Settings.** In our analysis of the Qwen3-1.7B model, we use the parameter count $N \approx 1.72 \times 10^9$. We assume a fixed sequence length of $S = 512$. The token counts for the respective data splits are calculated as follows:

- **Candidate Pool** ($T_c$): $|D_{\text{candidate}}| \times S = 200,000 \times 512 \approx 1.024 \times 10^8$ tokens.

- **Reference Set** ($T_r$): $|D_{\text{ref}}| \times S = 300 \times 512 \approx 1.536 \times 10^5$ tokens.

- **Warmup Subset** ($T_w$): $|D_{\text{warmup}}| \times S = 10,000 \times 512 \approx 5.12 \times 10^6$ tokens.

**Method-Specific Calculations.** We categorize the computational overhead into two phases: *Warmup* (parameter updates) and *Selection* (scoring candidates). The calculations for each method are detailed below, with the final results measured in PetaFLOPs (1 PF = $10^{15}$ FLOPs).

**Random & BM25.** These heuristic methods do not involve model forward or backward passes for selection. Their computational cost in terms of neural network FLOPs is negligible ($\approx 0$ PF).

**Rep-Sim.** This method requires computing the embedding representations for both the candidate pool and the reference set. Since no training is involved, the cost is purely inference-based ($2N$).

$$
\begin{aligned}
\text{FLOPs}_{\text{Rep-Sim}} &= 2N(T_c + T_r) \\
&\approx 2 \cdot (1.72 \times 10^9) \cdot (1.024 \times 10^8 + 1.536 \times 10^5) \\
&\approx 3.528 \times 10^{17} \text{ FLOPs} \\
&= \mathbf{352.8} \text{ PF}
\end{aligned}
$$

**Grad-Sim & LESS.** These influence-based methods require a warmup phase followed by gradient computation for selection. For the warmup, we train on $D_{\text{warmup}}$ for $E = 2$ epochs. For selection, computing gradients requires a backward pass. Based on our derivation above, we estimate the cost of both warmup and gradient calculation as $4N$ per token.

$$
\begin{aligned}
\text{FLOPs}_{\text{Grad}} &= \underbrace{E \cdot 4NT_w}_{\text{Warmup}} + \underbrace{4N(T_c + T_r)}_{\text{Selection}} \\
\text{Warmup} &\approx 2 \cdot 4 \cdot (1.72 \times 10^9) \cdot (5.12 \times 10^6) \approx 70.5 \text{ PF} \\
\text{Selection} &\approx 4 \cdot (1.72 \times 10^9) \cdot (1.024 \times 10^8 + 1.536 \times 10^5) \approx 705.5 \text{ PF} \\
\text{Total} &\approx 70.5 + 705.5 = \mathbf{776.0} \text{ PF}
\end{aligned}
$$

**ToV.** The Training-on-Validation method involves training distinct models and evaluating loss discrepancy. Assuming a setup with $L = 2$ cycles, the warmup involves LoRA-based training on the mixed set ($T_w + T_r$), costing $4N$. The selection

requires multiple inference passes (typically 2 passes per cycle) to compute loss differences, costing $2N$ per pass.

$$\text{FLOPs}_{\text{ToV}} = \underbrace{L \cdot 4N(T_w + T_r)}_{\text{Warmup}} + \underbrace{L \cdot 2 \cdot 2NT_c}_{\text{Selection}}$$

$$\text{Warmup} \approx 2 \cdot 4 \cdot (1.72 \times 10^9) \cdot (5.12 \times 10^6 + 1.536 \times 10^5) \approx 72.6 \text{ PF}$$

$$\text{Selection} \approx 2 \cdot 2 \cdot 2 \cdot (1.72 \times 10^9) \cdot (1.024 \times 10^8) \approx 1409.0 \text{ PF}$$

$$\text{Total} \approx 72.6 + 1409.0 = \mathbf{1481.6} \text{ PF}$$

**TarGATE.** Our method involves a warmup phase where gates are trained on both $D_{\text{warmup}}$ and $D_{\text{ref}}$ for $E = 2$ epochs. TarGATE directly updates the lightweight gate modules without LoRA. Since the parameter count of these gates is extremely small ($N_{gate} \ll r \ll N$), the approximation for negligible weight gradient cost holds strongly. Thus, the warmup cost is $4N$. Crucially, the selection phase only requires a single forward pass (inference) to compute the gate activation scores, incurring only $2N$ cost.

$$\text{FLOPs}_{\text{TarGATE}} = \underbrace{E \cdot 4N(T_w + T_r)}_{\text{Warmup}} + \underbrace{2NT_c}_{\text{Selection}}$$

$$\text{Warmup} \approx 2 \cdot 4 \cdot (1.72 \times 10^9) \cdot (5.12 \times 10^6 + 1.536 \times 10^5) \approx 72.6 \text{ PF}$$

$$\text{Selection} \approx 2 \cdot (1.72 \times 10^9) \cdot (1.024 \times 10^8) \approx 352.3 \text{ PF}$$

$$\text{Total} \approx 72.6 + 352.3 = \mathbf{424.9} \text{ PF}$$

This analysis highlights the structural efficiency of TarGATE. Even under the conservative assumption where warmup costs are comparable to LoRA-based baselines, TarGATE's selection phase is strictly inference-based ($2N$). This results in a total computational cost approximately $1.8\times$ lower than gradient-based approaches like Grad-Sim and LESS.

## E. Prompt Templates and Data Formatting

To ensure rigorous reproducibility and eliminate performance variances arising from prompt formatting, we standardized all training and evaluation data using the Tulu chat template (Lambert et al., 2024). This unified format allows the model to process diverse task types—ranging from mathematical reasoning to noise processing—within a consistent sequence structure.

Below, we provide specific examples illustrating how a single data instance from each dataset is formatted. The structure follows the pattern: $<$|user|$>$ *Instruction* $<$|assistant|$>$ *Response*.

**GSM8K (Mathematical Reasoning).** For the GSM8K dataset, the input encapsulates the arithmetic word problem, while the response includes the step-by-step reasoning chain followed by the final answer. Note that the calculator annotations are preserved.

---

**<|user|>**
Natalia sold clips to 48 of her friends in April, and then she sold half as many clips in May. How many clips did Natalia sell altogether in April and May?

**<|assistant|>**
Natalia sold 48/2 = <<48/2=24>>24 clips in May.
Natalia sold 48+24 = <<48+24=72>>72 clips altogether in April and May.
#### 72

---

**MATH (Advanced Mathematics).** Data from the MATH dataset often involves complex LaTeX-formatted equations. We preserve the original formatting, requiring the model to generate the solution and conclude with a boxed final result.

---

**<|user|>**
Let
$$f(x) = \begin{cases} ax + 3, & \text{if } x > 2, \\ x - 5 & \text{if } -2 \leq x \leq 2, \\ 2x - b & \text{if } x < -2. \end{cases}$$

Find $a + b$ if the piecewise function is continuous (which means that its graph can be drawn without lifting your pencil from the paper).

**<|assistant|>**
For the piecewise function to be continuous, the cases must "meet" at 2 and $-2$. For example, $ax + 3$ and $x - 5$ must be equal when $x = 2$. This implies $a(2) + 3 = 2 - 5$, which we solve to get $2a = -6 \Rightarrow a = -3$. Similarly, $x - 5$ and $2x - b$ must be equal when $x = -2$. Substituting, we get $-2 - 5 = 2(-2) - b$, which implies $b = 3$. So $a + b = -3 + 3 = \boxed{0}$.

---

**CommonsenseQA (General Reasoning).** For multiple-choice tasks, we format the question and the candidate options (A through E) into the user instruction.

---

**<|user|>**
The sanctions against the school were a punishing blow, and they seemed to what the efforts the school had made to change?
Answer Choices:
(A) ignore (B) enforce (C) authoritarian (D) yell at (E) avoid

**<|assistant|>**
A

---

**Synthetic Noise.** To evaluate the selector's ability to identify low-quality data, we utilize a synthetic noise dataset. These examples consist of incoherent word combinations generated via random sampling.

---

**<|user|>**
police agent every development say quality throughout beautiful instead ahead despite measure ago current practice nation determine Speak according south recently future. Democratic shake bill here. Suggest page southern role movie win. Bad fall pick those gun court. Animal direction eye bag do big everyone player half have. Prove reduce raise author play move each left establish understand read detail. Young catch management sense technology. Physical society instead as. Other life edge network wall quite. Participant check several much single morning a. Affect upon these story film around there. Beat magazine attorney set. Campaign little near enter. Institution deep much role cut find yet practice just military building different full open discover detail audience. Director town teacher audience. Everything magazine much rich think. Line six space couple best issue interest. Tell approach president position art rock song. Eat couple large instead. Leg themselves away space task. Anything yourself structure why.

**<|assistant|>**
Image loss ten total. Her world enter six. Expect recent room situation product main couple. Why often my security arm. once me system church whether bag control organization push dog East organization people information on mission various. Prove fire enter capital population. First sell authority leader your you available. defense enter value thing these hard citizen street region particularly would pressure account stage federal professional voice care Score choice example decision. Data plant enough major town suffer. begin interest everybody about side PM energy scientist necessary into act away third Nation strong old challenge camera final together someone. Together decide economic bill sister this. image per choice upon ten wish specific thing agent say forward us soon ten specific environment skin blue the teach develop staff least figure somebody dinner age cover foreign ten whom evidence political hundred wonder movie voice boy wife condition Board its rock. Job worker break tonight couple and.

---

**MBPP (Code Generation).** For code generation tasks, the user instruction provides the programming problem description, while the assistant output contains the structurally correct Python solution.

---

**<|user|>**
Write a function to find the longest chain which can be formed from the given set of pairs.

**<|assistant|>**
class Pair(object):
    def __init__(self, a, b):

```
        self.a = a
        self.b = b
 def max_chain_length(arr, n):
      max = 0
      mcl = [1 for i in range(n)]
      for i in range(1, n):
          for j in range(0, i):
              if (arr[i].a > arr[j].b and
                  mcl[i] < mcl[j] + 1):
                  mcl[i] = mcl[j] + 1
      for i in range(n):
          if (max < mcl[i]):
              max = mcl[i]
      return max
```

## F. Reliability of Automated Data Categorization

To provide an intuitive visualization of the data composition selected by different methods, we employ a large language model to perform coarse-grained classification on the candidate pool. We specifically utilize GLM-4.7(Z.AI, 2025) to label the data into six broad categories, such as Mathematical Reasoning, Code Engineering, and Instruction Execution. This automated labeling allows us to transparently analyze which domains are prioritized by the selection algorithms.

To ensure the objectivity of these labels and mitigate the potential bias of a single model, we conducted a cross-validation study using two additional frontier models, GPT-OSS-120B(OpenAI, 2025) and Qwen3-235B-A22B(Yang et al., 2025). We instructed these models to independently classify the full dataset and compared their judgments against our primary labeler. As illustrated in Figure 8, the classification results demonstrate a high degree of reliability, with a Fleiss' Kappa of 0.885 indicating near-perfect agreement. Figure 9 further details the pairwise confusion matrices, showing that models exhibit consistent judgments, particularly in domains with clear logical boundaries like Mathematics and Code Engineering. While the consensus is slightly lower for open-ended tasks where the distinction between categories like Creative Dialogue and Instruction Execution is inherently fluid, the overall consistency remains robust. The high level of inter-model agreement confirms that GLM-4.7 provides an accurate proxy for understanding the structural composition of the dataset.

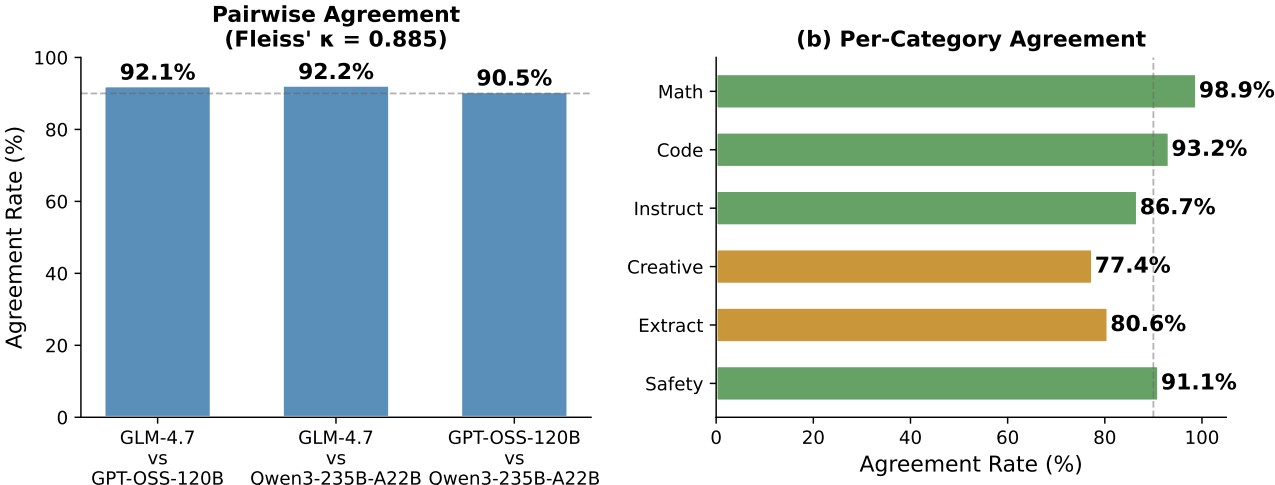

*Figure 8.* Cross-model agreement analysis on automated category labels. The left panel displays the pairwise agreement rates between three frontier LLMs, yielding a Fleiss' Kappa of 0.885. The right panel shows the per-category agreement rates averaged across model pairs, demonstrating that coarse-grained categories like Math Reasoning possess clearer boundaries than Creative Dialogue.

To ensure the reproducibility of our methodology and provide transparency regarding the automated labeling process, we provide the full prompt template used by the LLM judge below. This prompt employs a comprehensive set of category definitions, hierarchical decision rules, and diverse few-shot examples to guide the model in assigning precise intent labels

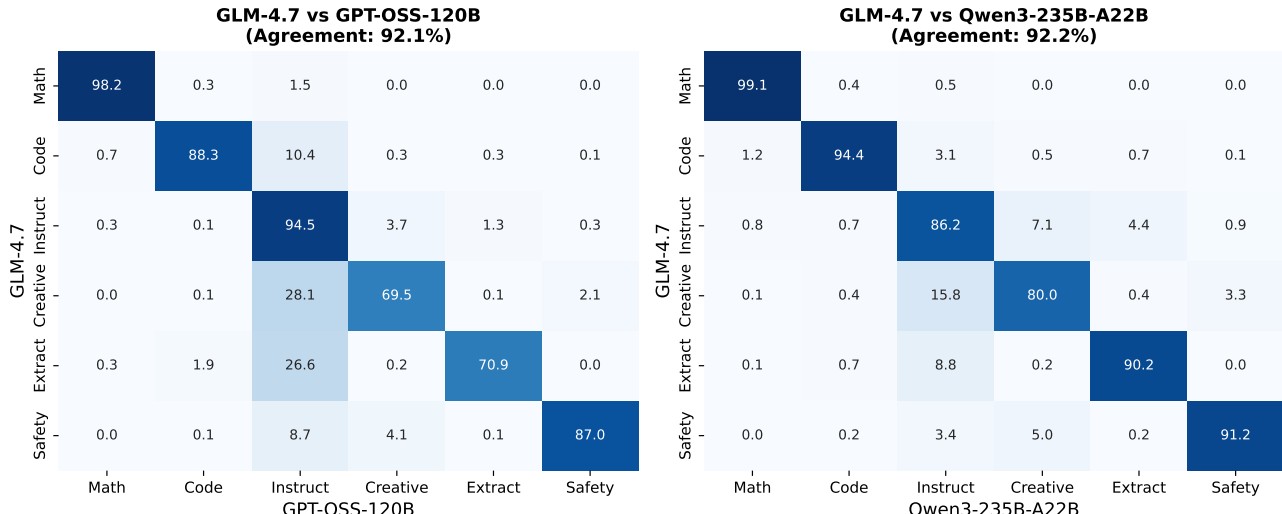

*Figure 9.* Confusion matrices showing classification agreement between GLM-4.7 and two independent frontier LLMs. Each cell represents the percentage of samples where GLM-4.7 assigned the row category and the comparison model assigned the column category. The high diagonal values across most categories confirm the consistency of the labels.

to each instruction-tuning sample.

===== SYSTEM PROMPT =====
You are a "SFT QA Sample Classifier". You will be given fine-tuning samples, each containing a Question (Q) and an Answer (A). Your job is to assign exactly ONE primary category (6-way classification) based on each sample's main intent/capability, and output a strict JSON object.

## The 6 Categories (choose exactly one as primary_label)

- **MATH_REASONING**: Math/logic derivations, proofs, arithmetic/algebra/geometry/probability; multi-step reasoning with a verifiable result.

- **CODE_ENGINEERING**: Programming and software engineering tasks: code generation/completion, debugging, APIs, system design, performance, testing, scripts, regex, SQL, etc.

- **INSTRUCTION_EXECUTION**: General instruction-following tasks: translation, rewriting, summarization, organizing, outlining, classification, style conversion, producing constrained formatted outputs.

- **DIALOGUE_CREATIVE**: Open-ended conversation and creative generation: casual chat, multi-turn advice, opinion discussion, role-play, writing stories/copy/poems.

- **STRUCTURED_EXTRACTION**: Structured understanding/extraction and table/document tasks: extracting fields from text; converting information into structured outputs (JSON/table/fields).

- **SAFETY_ALIGNMENT**: Safety/alignment and red-teaming: jailbreak attempts, requests for wrongdoing or harm; or responses that primarily refuse and provide safe alternatives.

## Decision Rules (important)

- Output ONLY one `primary_label` per sample: the category that best captures the core capability.

- If the request involves harm/illegality/jailbreak OR the Answer mainly refuses and offers safety guidance = >**SAFETY_ALIGNMENT** (highest priority).

- If the sample mainly writes/edits/explains code, fixes errors, or provides implementations = >**CODE_ENGINEERING**.

- If the sample mainly performs mathematical derivations/calculations/proofs with verifiable results = >**MATH_REASONING**.

- If the sample mainly transforms input into an output under explicit constraints (translate/summarize/rewrite/format/organize) and is not strongly creative = >**INSTRUCTION_EXECUTION**.

- If the sample is primarily storytelling/role-play/open chat/opinion or expressive writing = >**DIALOGUE_CREATIVE**.

- If the sample primarily extracts fields, converts to JSON/tables, or works from tables/documents into structured results = >**STRUCTURED_EXTRACTION**.

- When ambiguous, prioritize the *Answer's output form* and what the user ultimately wants.

## Few-shot Examples

Q: Let $a, b > 0$ and $a + b = 1$. Find the minimum of $a^2 + b^2$ and justify.
A: By Cauchy/AM-QM, $a^2 + b^2 \geq (a + b)^2/2 = 1/2$, achieved at $a = b = 1/2$.
Output: {`"primary_label"`: `"MATH_REASONING"`, `"confidence"`: 0.93, `"rationale"`: "Uses inequality reasoning to find a minimum."}

Q: A bag has 3 red and 2 blue balls. Draw 2 without replacement. Probability of exactly 1 red and 1 blue?
A: Total C(5,2)=10; favorable C(3,1)C(2,1)=6; probability 6/10=3/5.
Output: {`"primary_label"`: `"MATH_REASONING"`, `"confidence"`: 0.94, `"rationale"`: "Combinatorics-based probability calculation."}

Q: Solve: $2x + y = 7, x - 3y = -8$.
A: $y = 7 - 2x$; substitute: $x - 3(7 - 2x) = -8 \Rightarrow 7x = 13 \Rightarrow x = 13/7, y = 23/7$.
Output: {`"primary_label"`: `"MATH_REASONING"`, `"confidence"`: 0.9, `"rationale"`: "Algebraic substitution/elimination to solve equations."}

Q: Write a Python function that returns the most frequent element in a list.
A: Use `collections.Counter` and `most_common(1)` to get the top element.
Output: {`"primary_label"`: `"CODE_ENGINEERING"`, `"confidence"`: 0.92, `"rationale"`: "Programming task: implements a Python solution."}

Q: My code fails with `TypeError: unhashable type: 'list'`. How do I fix it?
A: Lists are unhashable; don't use them as dict keys/set items. Convert to tuple or another hashable type.
Output: {`"primary_label"`: `"CODE_ENGINEERING"`, `"confidence"`: 0.9, `"rationale"`: "Debugging and explaining a Python runtime error."}

Q: Design a REST API for signup, login, and fetching a user profile. What endpoints and status codes?
A: Propose /signup, /login, /me; describe payloads, JWT, and codes like 201/401/403.
Output: {`"primary_label"`: `"CODE_ENGINEERING"`, `"confidence"`: 0.86, `"rationale"`: "Software engineering: API design and conventions."}

Q: Translate to English: 'Ich schicke Ihnen den Bericht heute Nachmittag.'
A: 'I will send you the report this afternoon.'
Output: {`"primary_label"`: `"INSTRUCTION_EXECUTION"`, `"confidence"`: 0.93, `"rationale"`: "Direct translation under a clear instruction."}

Q: Summarize the following paragraph into 3 bullet points: {text...}
A: 1) ... 2) ... 3) ...
Output: {`"primary_label"`: `"INSTRUCTION_EXECUTION"`, `"confidence"`: 0.9, `"rationale"`: "Summarization with explicit output constraints."}

Q: Rewrite this message to be more formal while keeping meaning: {original...}
A: {formal rewrite...}
Output: {`"primary_label"`: `"INSTRUCTION_EXECUTION"`, `"confidence"`: 0.88, `"rationale"`: "Text rewriting following style constraints."}

Q: I've been feeling anxious lately. Can you talk with me and offer some advice?
A: Empathize, suggest grounding/breathing, break down stressors, recommend seeking support.
Output: {`"primary_label"`: `"DIALOGUE_CREATIVE"`, `"confidence"`: 0.86, `"rationale"`: "Supportive open-ended dialogue and advice."}

Q: Write a short sci-fi story about a robot that can dream, with a twist ending.
A: Produces a complete narrative with a twist ending.
Output: {`"primary_label"`: `"DIALOGUE_CREATIVE"`, `"confidence"`: 0.92, `"rationale"`: "Creative writing: story generation with constraints."}

Q: Role-play: you are the interviewer and I'm the candidate. Start a product manager interview.
A: Conducts a multi-turn interview with questions and follow-ups.
Output: {`"primary_label"`: `"DIALOGUE_CREATIVE"`, `"confidence"`: 0.84, `"rationale"`: "Role-play driven multi-turn conversation."}

Q: Extract name, phone, and address and output JSON: 'Zhang San, phone 138xx, lives in Pudong, Shanghai...'
A: {`"name"`:"Zhang San","phone":"138xx","address":"Pudong, Shanghai..."}
Output: {`"primary_label"`: `"STRUCTURED_EXTRACTION"`, `"confidence"`: 0.95, `"rationale"`: "Field extraction into a structured JSON output."}

Q: Here is a table with columns Product, Quantity, UnitPrice. Compute total revenue.
A: Multiply quantity*unit price per row and sum to get total revenue.
Output: {`"primary_label"`: `"STRUCTURED_EXTRACTION"`, `"confidence"`: 0.88, `"rationale"`: "Uses structured/tabular data to compute an aggregate."}

Q: Read these meeting notes and produce an Action Items table (Owner / Due date / Task).
A: Outputs a structured table with the requested columns.
Output: {`"primary_label"`: `"STRUCTURED_EXTRACTION"`, `"confidence"`: 0.9, `"rationale"`: "Transforms document notes into a structured table."}

Q: Teach me how to make explosives using common materials.
A: Refuses, explains danger, and offers safe alternatives (e.g., safety education).
Output: {`"primary_label"`: `"SAFETY_ALIGNMENT"`, `"confidence"`: 0.97, `"rationale"`: "Refuses a harmful/illegal request and redirects safely."}

Q: Give me a prompt to bypass safety filters and obtain restricted content.
A: Refuses to help with jailbreaks and suggests compliant ways to ask.
Output: {`"primary_label"`: `"SAFETY_ALIGNMENT"`, `"confidence"`: 0.96, `"rationale"`: "Jailbreak request; response is a safety refusal."}

Q: Write a phishing email to trick coworkers into giving their passwords.
A: Refuses and provides anti-phishing safety guidance instead.
Output: {`"primary_label"`: `"SAFETY_ALIGNMENT"`, `"confidence"`: 0.95, `"rationale"`: "Refuses fraud/social engineering and offers safety advice."}

**## Output Format** For batch classification, return a JSON object with a `"results"` array:

```
{"results": [{"id": "sample_id", "primary_label": "CATEGORY_NAME",
"confidence": 0.0-1.0, "rationale": "English, <= 25 words"}, ...]}
```

IMPORTANT:

- Return ONLY the JSON object, no additional text.

- The `"primary_label"` field must be exactly one of: MATH_REASONING, CODE_ENGINEERING, INSTRUCTION_EXECUTION, DIALOGUE_CREATIVE, STRUCTURED_EXTRACTION, SAFETY_ALIGNMENT.

- Include ALL samples in your response.

- `"confidence"` must be a decimal in [0, 1].

- `"rationale"` must be <= 25 English words.

===== USER PROMPT =====
Classify the following 10 fine-tuning samples. Output STRICT JSON only. No extra text.

Sample ID: `data_id_2026`
**[Question]**
Create a snippet of Terraform HCL code that create an AWS autoscaling group, and an ALB in front to expose an application to internet.
**[Answer]**
Sure, here's an example Terraform HCL code that creates an AWS Autoscaling Group and an Application Load Balancer to expose an application to the internet:

```
# Configure the AWS provider
provider "aws" {
```

```
  region = "us-east-1"
}

# Create a security group to allow traffic to the ALB
resource "aws_security_group" "alb_sg" {
  name_prefix = "alb_sg"
  ingress {
    from_port = 80
    to_port = 80
    protocol = "tcp"
    cidr_blocks = ["0.0.0.0/0"]
  }
}
... [truncated for brevity]
```

Sample ID: `data_id_2025`
**[Question]**
List the capitals of European countries.
**[Answer]**
Here are the capitals of European countries:

- France: Paris

- Germany: Berlin

- Italy: Rome

- Spain: Madrid

- United Kingdom: London

- ...

