# OpenReview forum: "TarGATE: Target-Aware Data Selection via Token-Attenuation Gates"
_ICML.cc/2026/Conference — ICML 2026 regular_

### Official Review · Reviewer_niJ7 · 2026-02-27

**Soundness:** 3
**Presentation:** 4
**Significance:** 3
**Originality:** 3
**Overall Recommendation:** 4
**Confidence:** 4

**Summary:**

This paper proposes TarGate, a data selection method designed to extract a specialized subset from a large candidate SFT dataset for specific target tasks. The core idea is to introduce learnable gates that attenuate the outputs of FFNs within transformer blocks. Each gate produces a scalar activation between 0 and 1, referred to as the Information Retention Ratio (IRR), which serves as a measure of token quality. The quality score of a sequence is defined as the average IRR across all its tokens.

To obtain IRR scores, the LLM is frozen, and only the gating parameters are trained. The gates are optimized using the standard SFT loss, combined with an auxiliary regularization objectives that encourage IRR sparsity on samples from the candidate dataset while discouraging IRR sparsity on a reference dataset corresponding to the target task. This contrastive sparsity design enables TarGate to distinguish task-relevant data from less relevant samples.

Experiments on both proof-of-concept and real-world datasets demonstrate the effectiveness of TarGate. As for the computational efficiency, TarGate achieves better computational efficiency compared to gradient-based data selection baselines, but it is slightly less efficient than embedding-based methods.

**Compliance With Llm Reviewing Policy:**

Affirmed.

**Final Justification:**

My concerns are addressed, and I will keep my positive rating.

**Key Questions For Authors:**

1. In the *Implementation Details* (Sec. 4.1), it is unclear what the reported random seeds refer to. Do they correspond only to the seeds used during fine-tuning, or do they also include the seeds used for random data selection (baseline) and the random warm-up data selection from the candidate set in TarGate?

If the reported seeds apply only to SFT, I suggest that the authors clarify whether randomness in the data selection process could affect the conclusions.

2. See other questions in Weaknesses.

I will be happy to increase my rating if the authors resolve my concerns.

**Limitations:**

This work does not discuss limitations.

**Strengths And Weaknesses:**

## Strengths:

1. The idea of training gates and using gating activations to measure token- and sequence-level quality is both interesting and conceptually simple (in a positive sense).

2. The design introduces only a very small number of additional trainable parameters, which is important for efficiency and scalability.

3. The experiments clearly demonstrate the effectiveness of the method. The inclusion of a proof-of-concept dataset is particularly helpful and allows readers to more directly observe and understand the benefits of the approach.

4. The analyses in Sections 4.3 and 4.4 present the effectiveness of the proposed method in an intuitive and interpretable way.

5. The paper is clearly written and easy to follow.

## Weaknesses:

1. The evaluation only includes three target benchmarks, which is somewhat limited. I suggest the authors to expand the evaluation to include additional benchmarks like MMLU, TyDiQA, and BBH, especially since these benchmarks are evaluated in related works such as LESS and DATE-LM.

2. The paper lacks comparisons with general-purpose SFT data selection methods that do not rely on reference datasets. Such methods are often more computationally efficient and aim to improve performance across a broad range of downstream tasks.
Although the current work focuses on task-specific data selection, it would still be valuable to compare against general-purpose selection methods, and discuss the trade-offs between task-specific and general-purpose approaches.
Some (non-exhaustive) general-purpose data selection baselines include:
   * Heuristic-based: Longest response length [1]
   * Perplexity-based: IFD [2]
   * Perplexity-based: T-SHIRT [3] (which also has a token-selective flavor)

   There is also a baseline that uses a reference dataset from the target task:
      * Shapley value-based: SHED [4] (similar to this work, it relies on reference datasets for target tasks)

3. The paper claims that $\mathcal{L}_{\text{warmup}}$ is used to filter general low-quality noise (L218–219). One could imagine that, by setting $\beta = 2$ in Eq. (7), TarGate may function as a general-purpose data selection method.
However, the raw values (transparent lines) in Fig. 5 at $\beta = 2$ suggest that TarGate performs no better than, or even worse than, the random selection baseline on three benchmarks. This raises concerns regarding:
   * The generalization ability of TarGate;
   * The role of $\mathcal{L}_{\text{warmup}}$;

   because people often aim to improve performance across a suite of tasks rather than a single task in the real world.

## Minor Weaknesses:

4. The discussion in Section 2.3 is somewhat coarse-grained. There has been substantial recent work on gating mechanisms in Transformers (e.g., recent NeurIPS 2025 best paper on gated attention [5]).

5. The main paper misses lots of cross-references to the appendix, which makes it harder to find more details in the appendix.

**References**:

[1] Long Is More for Alignment: A Simple but Tough-to-Beat Baseline for Instruction Fine-Tuning, ICML 2024.

[2] From Quantity to Quality: Boosting LLM Performance with Self-Guided Data Selection for Instruction Tuning, NAACL 2024.

[3] T-SHIRT: Token-Selective Hierarchical Data Selection for Instruction Tuning, NeurIPS 2025.

[4] SHED: Shapley-Based Automated Dataset Refinement for Instruction Fine-Tuning, NeurIPS 2024.

[5] Gated Attention for Large Language Models: Non-linearity, Sparsity, and Attention-Sink-Free, NeurIPS 2025

---

> ### Author Rebuttal · Authors · 2026-03-31
>
> Hi, Reviewer niJ7:
>
> Thank you for the careful reading and detailed suggestions. Your main concerns focus on broader evaluation, baseline coverage, the joint objective, and clarity issues. Unless noted, added experiments follow the original main setting. We summarize the results below.
>
> ***`Q1: "Clarify what randomness the reported seeds cover"`***
>
> **R1:** Thank you for pointing this out. In our experiments, the reported seed is fixed for the **entire pipeline**, from the warmup stage through the final fine-tuning stage; i.e., each run uses one seed consistently rather than reseeding different stages separately. We will clarify this in the revision.
>
> ***`W1: "Broader evaluation is needed beyond the current three target benchmarks"`***
>
> **R2:** Under Qwen3-0.6B -> Qwen3-1.7B, the original and newly added task sets are:
>
> |Task set|Tasks|TarGATE scores|
> |---|---|---|
> |paper|CQA / GSM8K / MBPP|73.7 / 62.1 / 57.2|
> |new|MMLU / BBH / TyDiQA|60.9 / 53.6 / 68.6|
>
> New comparisons:
>
> |Method|MMLU|BBH|TyDiQA|
> |---|---:|---:|---:|
> |TarGATE|**60.9**|**53.6**|**68.6**|
> |BM25|60.7|53.4|68.4|
> |Grad-Sim|60.7|52.8|67.9|
> |LESS|60.7|**53.6**|68.0|
> |Random|60.5|53.0|68.1|
> |RDS+|60.6|51.6|67.6|
> |Rep-Sim|60.7|52.8|68.0|
>
> The gains are smaller than on GSM8K/MBPP, but they remain consistent or tied-best. We view this as useful supporting evidence: **TarGATE remains competitive on the added benchmarks**, and **the advantage is not restricted to one narrow task type**. We agree that still broader task coverage would be valuable, and we will note this as future work.
>
> ***`W2: "Missing comparisons with general-purpose or alternative reference-based baselines"`***
>
> **R3:** We added the requested broader baseline set: the general-purpose methods Longest [1], IFD [2], and T-SHIRT [3], the reference-based SHED [4], and also MONA [6]. We implemented MONA under the same Qwen3-0.6B selector + Qwen3-1.7B target + top-5% selection setting, instantiating the missing Qwen3-0.6B SAE component with a RAGLens [7]-style recipe and keeping the remaining MONA pipeline aligned with the original method. We place them together with the original major baselines:
>
> |Method|CQA|GSM8K|MBPP|
> |---|---:|---:|---:|
> |Random|72.7|58.8|54.0|
> |BM25|72.8|57.7|55.2|
> |Grad-Sim|72.4|59.6|54.8|
> |LESS|73.0|58.6|54.8|
> |Rep-Sim|72.7|60.6|52.2|
> |RDS+|73.0|61.9|55.2|
> |Longest|70.8|37.7|54.8|
> |IFD|71.7|52.2|55.4|
> |T-SHIRT|71.4|36.9|55.2|
> |SHED|72.1|58.4|54.2|
> |MONA|72.8|61.1|55.2|
> |TarGATE|**73.7**|**62.1**|**57.2**|
>
> These results strengthen the comparison to both general-purpose and reference-based alternatives. The main takeaway is that **TarGATE remains much stronger than general-purpose selectors on target-specific tasks**, especially GSM8K, and **also outperforms SHED [4] and MONA [6]**. We agree that continued expansion of baseline coverage is worthwhile, and we will reflect this in the revision.
>
> ***`W3: "Clarify the behavior in the degenerate general-purpose regime"`***
>
> **R4:** Our existing beta ablations already show that **both objectives are needed**: boundary settings where only one objective remains are clearly worse. This shows TarGATE does not work by only memorizing the reference signal or only denoising the candidate pool. Rather, it benefits from the contrast between the target-aligned reference set and the broader candidate distribution. The new general-purpose baselines support the same conclusion from another angle: on GSM8K, TarGATE reaches 62.1, compared with 52.2 for IFD, 36.9 for T-SHIRT, and 37.7 for Longest. This is consistent with our intended use case: **TarGATE is a target-aware selector, not a general-purpose filter**. We agree that this boundary regime is still worth deeper study.
>
> ***`W4: "Section 2.3 is too coarse-grained on recent gating work"`***
>
> **R5:** You are right that Sec. 2.3 is too coarse-grained. This is a weakness of our presentation, because it can blur how TarGATE relates to recent Transformer gating work. In the revision, we will expand Sec. 2.3 to discuss work like gated attention [5], while clarifying that TarGATE uses gates as a data-centric probe rather than as an attention-layer redesign.
>
> ***`W5: "The main paper misses appendix cross-references"`***
>
> **R6:** We apologize that the main paper currently under-links the appendix. This is a presentation issue rather than a lack of detail, since several implementation and baseline details are already in the appendix but are harder to find than they should be. We will add explicit appendix pointers throughout the main paper and make the limitations discussion more explicit in the revision.
>
> Refs: [6] Ma et al. Task-Specific Data Selection via Monosemantic Neuronal Activations. NeurIPS 2025. [7] Xiong et al. Toward Faithful Retrieval-Augmented Generation with Sparse Autoencoders. ICLR 2026.
>
> **Thank you again for your valuable feedback, which has greatly improved our work. We hope these responses address your concerns, and we would be happy to clarify any remaining questions.**

---

> > ### Author Rebuttal · Reviewer_niJ7 · 2026-03-31
> >
> > I would like to thank the authors for their response. Most of my concerns have been addressed, with the exception of W3. While I understand that the scope of the paper focuses on targeted selection, I place higher value on approaches that have the potential to support a suite of downstream tasks rather than a single one, as it is more common in SFT to aim for broader applicability.
> >
> > I would like to raise my rating to 4.

---

> > > ### Author Response · Authors · 2026-04-03
> > >
> > > Thank you for raising this important remaining question, and for your constructive feedback throughout the review process. We agree that, in many SFT settings, it is more useful to select data that can support several downstream tasks, rather than optimizing for only one target task.
> > >
> > > ***`Q2: "How does TarGATE perform in a multi-task setting?"`***
> > >
> > > **Comment:** Thank you for raising this important point. We understand your remaining concern as asking whether TarGATE can remain effective when the goal is no longer a single target task, but a multi-task setting. To directly answer this, we added a **multi-task selection** experiment under the Real-World Massive Scenario, using Qwen3-0.6B as the selector, Qwen3-1.7B as the target model, and the same top-5\% selection ratio as in our main experiments. Concretely, we mix the three task-specific reference sets (CommonsenseQA, GSM8K, MBPP) into one composite reference, run a single selection pass, fine-tune once on the selected subset, and then evaluate the resulting model on all three tasks. The results are as follows:
> > >
> > > |Method|CQA|GSM8K|MBPP|Avg|
> > > |---|---:|---:|---:|---:|
> > > |Random|72.7|58.8|54.0|61.8|
> > > |BM25|72.9|59.7|**55.6**|62.7|
> > > |Grad-Sim|73.0|58.4|53.8|61.7|
> > > |LESS|72.6|59.5|54.2|62.1|
> > > |Rep-Sim|72.7|**61.5**|55.0|63.1|
> > > |RDS+|72.4|60.8|54.6|62.6|
> > > |TarGATE|**73.4**|61.2|55.0|**63.2**|
> > >
> > > These results are particularly encouraging for the broader-use case you asked about. Under this more challenging setting, where the reference signal combines three target capabilities rather than focusing on a single task, TarGATE still achieves the best average performance overall. It is strongest on CQA and remains competitive on GSM8K and MBPP.
> > >
> > > More importantly, this result shows that TarGATE is not restricted to the single-task setting: a single subset selected from a mixed reference can already support multiple downstream tasks well. We therefore believe these results provide initial evidence that TarGATE has the potential to extend beyond the single-task setting studied in this paper toward broader SFT scenarios. We will add this experiment and discuss it explicitly in the revision. We also agree that this is an important direction for future work, and we will further study how to improve TarGATE in broader multi-task settings so that it can better adapt to scenarios with more diverse target-task requirements.
> > >
> > > Thank you again for highlighting this issue. We believe this additional evidence answers your concern more directly, and we will make this point explicit in the revision.

---

### Official Review · Reviewer_L8XZ · 2026-02-28

**Soundness:** 3
**Presentation:** 3
**Significance:** 3
**Originality:** 2
**Overall Recommendation:** 4
**Confidence:** 4

**Summary:**

In each layer, IRR is computed for each token using a simple linear transformation followed by a sigmoid function. The resulting IRR is then used as a gating value on the FFN output. During training, for D_warmup​, the model jointly optimizes the standard LM loss and a regularization term on the average IRR values, so that the FFN outputs are attenuated for noisy samples that contribute little to the LM loss. For D_ref​, the regularization term on the average IRR is applied in the opposite direction, in order to maximize target-specific information flow. After training the gating module, the IRR averaged over tokens and layers is used as the criterion for data selection. Across a range of language models, fine-tuning on coresets selected by TarGATE consistently outperforms prior coreset selection approaches.

**Compliance With Llm Reviewing Policy:**

Affirmed.

**Final Justification:**

My concerns about the weaknesses have been resolved, and I will maintain my positive score.

**Key Questions For Authors:**

Same as described in weakness.

**Limitations:**

yes

**Strengths And Weaknesses:**

strength
- New perspective on gating function: The gating-function idea is widely used, but leveraging the gate values themselves as a data-selection criterion is novel.
- Technical soundness: The proposed method is simple and cost-efficient, while it still outperforms other baselines.

weakness
- Hyperparameter $\beta$ sensitivity: As shown in Fig. 5, the most preferred $\beta$ value varies across datasets, and performance is highly sensitive to the choice of $\beta$. It is quite cumbersome to tune it every time we apply this method. Is there any guideline for setting $\beta$ in a dataset-dependent manner?
- Insufficient review and experimental baselines on task-specific data selection methods [1,2,3]: TarGATE gates the entire FFN output at each layer with a single IRR value, whereas [1] performs sparse autoencoding at the level of individual activations within a layer, which can be seen as a more fine-grained form of gating. Moreover, [1] is train-free, making it highly cost-efficient. Although they share the same gating philosophy, is TarGATE better than [1] in terms of performance and computational efficiency?

[1] Ma, Da, et al. "Task-Specific Data Selection for Instruction Tuning via Monosemantic Neuronal Activations." NeurIPS 2025.

[2] Liu, Zifan, et al. "TSDS: Data selection for task-specific model finetuning." NeurIPS 2024.

[3] Wu, Yang, et al. "Rose: A reward-oriented data selection framework for llm task-specific instruction tuning." EMNLP 2025 findings.

---

> ### Author Rebuttal · Authors · 2026-03-31
>
> Hi, Reviewer L8XZ:
>
> Thank you for recognizing the novelty and technical soundness of the method. We note that your main concerns focus on hyperparameter sensitivity and comparison to recent task-specific selection methods. Unless otherwise specified, all settings below follow the original main experiments. To make the rebuttal self-contained, we summarize the most relevant original and new results together below.
>
> ***`W1: "Hyperparameter sensitivity"`***
>
> **R1:** We already include a dedicated ablation on beta and lambda with Qwen3-0.6B as selector and Qwen3-1.7B as target. The main findings are stable and interpretable: CQA prefers larger beta, while GSM8K and MBPP prefer smaller beta; reducing the reference-set size shifts the optimal beta lower, which is consistent with the intuition that weaker target signal requires stricter filtering. For lambda, performance remains stable around 0.5-1.0 and degrades mainly at extreme values, so **lambda=1 is a robust default** rather than a brittle tuned point. In other words, the hyperparameters exhibit **clear task-dependent trends rather than arbitrary instability**. We agree that more principled hyperparameter selection would be valuable, and we will include this as future work.
>
> ***`W2: "Missing baselines on task-specific selection methods"`***
>
> **R2:** We added a direct comparison to MONA [1] under the same setting: Qwen3-0.6B selector, Qwen3-1.7B target, and top-5% selection. Because there is no public and convincing Qwen3-0.6B SAE checkpoint, we used the RAGLens [4] Qwen3-0.6B SAE recipe to instantiate the missing SAE component, while keeping the rest of the pipeline as close to MONA as possible: penultimate-layer activations, SAE-based monosemantic features, mean token pooling, target-prototype averaging, generalized Jaccard scoring, and the same downstream fine-tuning recipe as the other methods. We report it together with the original major baselines:
>
> |Method|CQA|GSM8K|MBPP|
> |---|---:|---:|---:|
> |Random|72.7|58.8|54.0|
> |BM25|72.8|57.7|55.2|
> |Grad-Sim|72.4|59.6|54.8|
> |LESS|73.0|58.6|54.8|
> |Rep-Sim|72.7|60.6|52.2|
> |RDS+|73.0|61.9|55.2|
> |MONA|72.8|61.1|55.2|
> |TarGATE|**73.7**|**62.1**|**57.2**|
>
> The result is straightforward: **TarGATE outperforms MONA on all three tasks**, by +0.9/+1.0/+2.0 on CQA/GSM8K/MBPP. Since MONA is the closest task-specific selector to TarGATE in mechanism among the methods you mentioned, this is the most direct empirical comparison. For efficiency, all wall-clock times were measured on a single RTX 4090: MONA's data selection took about 1.5h, while TarGATE's warmup plus data scoring took about 2h. MONA, however, also requires training the Qwen3-0.6B SAE, which took about 20h and is its main extra cost. Therefore, under a practical end-to-end accounting, **MONA is substantially more expensive than TarGATE in our setting**. We spell out the MONA implementation because, in the Qwen3-0.6B setting, how the SAE is obtained directly affects fairness and reproducibility. We will make this setup clearer in the revision and continue expanding comparisons to other recent task-specific selectors.
>
> ***`W3: "Related-work positioning should be sharpened"`***
>
> **R3:** We agree, and thank you for pointing out that our current positioning is not yet sharp enough. More precisely, MONA [1], TSDS [2], ROSE [3], and TarGATE are all task-specific selectors guided by a small target set, but they place target-aware signal in different parts of the pipeline. TSDS [2] performs distribution alignment in an external representation space via optimal transport and diversity regularization. ROSE [3] also remains an external scoring pipeline, but replaces next-token-loss alignment with reward-oriented influence estimation. MONA [1] is the closest to TarGATE because it also leverages internal activations, yet it still builds an SAE-based monosemantic feature space and performs generalized Jaccard matching in that space. By contrast, **TarGATE does not construct a separate similarity/transport/influence module; it learns target awareness directly inside the selector through lightweight gates, and uses forward-pass IRR itself as the score.** In this sense, MONA is better viewed as post-hoc matching over internal activation features, whereas TarGATE is a direct form of model-internal scoring. We will rewrite the related-work discussion along this axis in the revision so that both the closeness to MONA and the distinction from TSDS/ROSE are stated more clearly.
>
> Reference: [4] Xiong et al. Toward Faithful Retrieval-Augmented Generation with Sparse Autoencoders. ICLR 2026.
>
> **Thank you again for your valuable feedback, which has greatly improved our work. We hope these responses address your concerns, and we would be happy to clarify any remaining questions.**

---

> > ### Author Rebuttal · Reviewer_L8XZ · 2026-04-02
> >
> > Thank you for the effort you put into the additional experiments. All of my concerns about the weaknesses have been resolved, and I will maintain my score.

---

> > > ### Author Response · Authors · 2026-04-03
> > >
> > > We sincerely thank you for acknowledging that your concerns have been resolved and for the constructive feedback that significantly improved our manuscript. We deeply appreciate your time, expertise, and supportive evaluation of our work.

---

### Official Review · Reviewer_XzkG · 2026-03-11

**Soundness:** 3
**Presentation:** 3
**Significance:** 3
**Originality:** 3
**Overall Recommendation:** 4
**Confidence:** 3

**Summary:**

The paper proposes TarGATE, an efficient data selection framework for targeted instruction tuning. It introduces a lightweight token-attenuation gate before the FFN layers to compute an Information Retention Ratio (IRR), dynamically modulating information flow without relying on computationally expensive gradient tracing or static representation similarities. The gates are optimized via a dual-objective warmup phase to contrast noisy candidate data against target-aligned reference data. During selection, TarGATE scores and filters high-quality samples using a single forward pass. Experiments on large-scale datasets demonstrate that it outperforms state-of-the-art baselines, exhibiting strong computational efficiency and cross-model transferability.

**Compliance With Llm Reviewing Policy:**

Affirmed.

**Final Justification:**

My concerns have been addressed. I would like to raise my rating to 4.

**Key Questions For Authors:**

See weaknesses.

**Limitations:**

yes

**Strengths And Weaknesses:**

## Strengths
1. The proposed method is interesting, simple, and effective: TarGATE introduces an elegant token-attenuation gate (IRR) to filter data, avoiding the high complexity of gradient-based or representation-based approximations.

2. The paper is well written and easy to follow: The paper is well-structured, and the visualizations effectively illustrate the underlying selection dynamics.

3. The empirical evaluation is comprehensive: The paper studies both synthetic noisy settings and large-scale real-world settings, and compares against heuristic, embedding-based, and gradient-based baselines, which makes the experimental section convincing.

## Weaknesses
1. Limited diversity of selector models: The paper evaluates TarGATE with only two selector backbones. While the results are encouraging, the evidence is still somewhat limited for establishing robustness across different model families and scales. Adding a few additional selectors would make the empirical claim substantially stronger.

2. Cross-model generalization is only partially validated: the target models still come from only two families, and this may not be sufficient to fully establish the generality of the proposed selector across architectures. Additional results on a few more models would strengthen the claim of cross-model transferability.

3. The task scope is still relatively narrow: The evaluation mainly focuses on three capabilities: commonsense reasoning, mathematical reasoning, and code generation. Although these are meaningful testbeds, they do not fully cover other practically important instruction-tuning settings such as truthfulness, instruction following, or broader alignment-related data. Including a few additional tasks or datasets in these directions would improve the paper’s scope and make the conclusions more convincing.

**I fully understand the time constraints of the rebuttal period. For my concerns, it would already be helpful if the authors could provide a modest amount of additional evidence, e.g., experiments on a few lightweight models such as Phi or Gemma, and possibly one or two additional small-scale datasets.**

---

> ### Author Rebuttal · Authors · 2026-03-31
>
> Hi, Reviewer XzkG:
>
> Thank you for the positive assessment of our method, writing, and empirical study. We note that your main concerns focus on selector diversity, cross-model transfer, and task coverage. Following your suggestion, we added a targeted set of new experiments centered on lightweight extra models and a few additional benchmarks. Unless otherwise specified, all added experiments follow the original main real-world setting. To make the rebuttal self-contained, we also place the most relevant original results together with the new ones below.
>
> ***`W1: "Limited diversity of selector models"`***
>
> **R1:** We first summarize TarGATE under both the original and new selectors, all evaluated on Qwen3-1.7B:
>
> |Task|Qwen3-0.6B|Qwen3-1.7B|Gemma-3-1B-PT|Llama-3.2-1B|
> |---|---:|---:|---:|---:|
> |CQA|73.7|**74.1**|73.1|72.6|
> |GSM8K|**62.1**|61.1|61.3|60.6|
> |MBPP|**57.2**|56.4|56.8|56.4|
>
> For the two new selectors, the full comparisons are:
>
> |Method|Gemma-CQA|Gemma-GSM8K|Gemma-MBPP|Llama-CQA|Llama-GSM8K|Llama-MBPP|
> |---|---:|---:|---:|---:|---:|---:|
> |Grad-Sim|72.2|60.4|56.2|72.1|55.8|55.2|
> |LESS|72.6|59.5|56.6|72.3|57.7|54.0|
> |RDS+|72.3|58.4|56.4|**72.6**|58.3|56.0|
> |Rep-Sim|72.4|59.8|53.6|72.3|58.7|55.6|
> |TarGATE|**73.1**|**61.3**|**56.8**|**72.6**|**60.6**|**56.4**|
>
> Taken together, TarGATE remains **best across all four selectors** and does not rely on one specific backbone. This is exactly what we hoped to test with these additions: Gemma-3-1B-PT brings in a non-Qwen family, while Llama-3.2-1B checks whether the same pattern persists in another compact backbone. We attribute this to the fact that TarGATE still makes substantial use of the backbone model's data understanding ability. We also agree that broader selector diversity remains important, since it would help readers better understand TarGATE's generalization, and we will add more selector results in the final version.
>
> ***`W2: "Cross-model transfer is only partially validated"`***
>
> **R2:** We also summarize TarGATE across both the original and new target models under the same Qwen3-0.6B selector:
>
> |Task|Qwen3-1.7B|Qwen3-14B|Llama3-8B|Gemma-3-4B-PT|Phi-4-mini-instruct|
> |---|---:|---:|---:|---:|---:|
> |CQA|73.7|87.8|65.8|71.3|73.4|
> |GSM8K|62.1|84.4|58.2|46.7|74.5|
> |MBPP|57.2|73.8|51.0|50.4|56.0|
>
> For the two new target models, the full comparisons are:
>
> |Method|Gemma-CQA|Gemma-GSM8K|Gemma-MBPP|Phi-CQA|Phi-GSM8K|Phi-MBPP|
> |---|---:|---:|---:|---:|---:|---:|
> |BM25|70.6|44.5|49.2|72.3|73.8|53.4|
> |Grad-Sim|70.7|45.2|48.6|72.8|73.5|53.6|
> |LESS|71.1|44.9|49.8|71.9|74.1|53.6|
> |Random|70.9|45.3|49.4|71.6|74.0|54.6|
> |RDS+|70.4|46.0|47.4|72.0|71.9|55.4|
> |Rep-Sim|70.8|45.4|48.8|72.5|73.2|54.8|
> |TarGATE|**71.3**|**46.7**|**50.4**|**73.4**|**74.5**|**56.0**|
>
> Taken together, the original Qwen/Llama targets and the new Gemma/Phi targets show that **the same selector transfers across multiple target families**. This extends the original evidence beyond the two families in the submission, so the transfer claim is no longer tied only to the original Qwen/Llama pair. The gains in the new results remain stable rather than isolated wins. We agree with your concern that larger-scale transfer evidence would be more convincing, and we will add more target-model results in the final paper.
>
> ***`W3: "The task scope is still relatively narrow"`***
>
> **R3:** To make the expanded scope explicit, we summarize the original and new task sets together under Qwen3-0.6B -> Qwen3-1.7B:
>
> |Task set|Tasks|TarGATE scores|
> |---|---|---|
> |paper|CQA / GSM8K / MBPP|73.7 / 62.1 / 57.2|
> |new|MMLU / BBH / TyDiQA|60.9 / 53.6 / 68.6|
>
> The full new task comparisons are:
>
> |Method|MMLU|BBH|TyDiQA|
> |---|---:|---:|---:|
> |TarGATE|**60.9**|**53.6**|**68.6**|
> |BM25|60.7|53.4|68.4|
> |Grad-Sim|60.7|52.8|67.9|
> |LESS|60.7|**53.6**|68.0|
> |Random|60.5|53.0|68.1|
> |RDS+|60.6|51.6|67.6|
> |Rep-Sim|60.7|52.8|68.0|
>
> TarGATE's gains on these added tasks are smaller than on GSM8K or MBPP, but they remain **consistently favorable** or tied-best. We believe these results are still informative because they show that **TarGATE remains effective beyond the original commonsense/math/code tasks**, and stays competitive across multiple tasks. At the same time, we agree that these additions do not yet exhaust directions such as truthfulness or instruction following raised in your review; rather, they provide concrete extra evidence under rebuttal constraints and broaden the scope beyond the original three capability axes. We agree that expanding target-task coverage is worthwhile, and we will add more such experiments in the final version.
>
> **Thank you again for your valuable feedback, which has greatly improved our work. We hope these responses address your concerns, and we would be happy to clarify any remaining questions.**

---

> > ### Author Rebuttal · Reviewer_XzkG · 2026-04-01
> >
> > Thank you for your response. My concerns have been addressed. I would like to raise my rating to 4.

---

> > > ### Author Response · Authors · 2026-04-02
> > >
> > > Thank you very much for your positive update. We sincerely appreciate your time and the valuable suggestions you provided for our work.

---

### Official Review · Reviewer_Jq1L · 2026-03-13

**Soundness:** 3
**Presentation:** 3
**Significance:** 3
**Originality:** 3
**Overall Recommendation:** 4
**Confidence:** 5

**Summary:**

This paper proposes TarGATE, a target-aware data selection method for instruction tuning. It introduces a simple and lightweight gating mechanism for transformer decoder blocks. The gates take attention outputs as input and compute an Information Retention Ratio (IRR), which serves as a quality score for individual training instances. The gating module is trained through a joint optimization strategy over candidate data and a reference dataset, with the goal of better aligning the selected data with the reference distribution. After training, the resulting IRR scores are used to rank and select data. The method is simple and computationally efficient, requiring only a single forward pass for scoring after gate training. The empirical results are also strong across both synthetic and real-world settings.

**Compliance With Llm Reviewing Policy:**

Affirmed.

**Final Justification:**

The rebuttal has satisfactorily addressed my main concerns, and I now lean toward acceptance.

**Key Questions For Authors:**

- How sensitive is the method to the reference set quality and size?

- The final sample score is just the global average IRR, and the paper does not compare this design with other aggregation choices in depth. Why is the current IRR aggregation the best scoring choice?

- Can the authors provide more analysis of what kinds of samples TarGATE tends to select, beyond overall performance?

- Since IRR is a token-level signal, the paper would benefit from a clearer analysis of which tokens tend to receive high or low gate values. For example, it would be helpful to show whether the mechanism emphasizes reasoning-relevant or answer-related tokens while down-weighting boilerplate or less informative context. This would improve interpretability and help explain why the method works.

- The paper acknowledges that the 0.6B selector can slightly outperform the 1.7B selector, but the current explanation is not fully convincing. Since this is a somewhat counterintuitive result, more direct analysis would be helpful to support the claim.

**Limitations:**

The paper fail to explicitly discuss its limitations. The authors should comment on sensitivity to reference-set quality and size, the lack of token-level analysis of the gating behavior, and the limited explanation of the selector-scale effect.

**Strengths And Weaknesses:**

- The proposed method is technically reasonable, and the experimental setup, including the choice of baselines, is generally appropriate. The paper evaluates the method in both real-world and synthetic settings, which helps support its empirical claims and contributions. However, the paper provides limited analysis of the method’s robustness to the quality and size of the reference set.

- The paper is generally well-written and easy to follow.

- The paper addresses a practical efficiency problem in task-aware data selection compared with existing methods. Its main strengths are the simplicity of the approach, its efficiency relative to gradient-based alternatives, and its consistent empirical gains.

- The main novelty lies in using lightweight token-level gates as a target-aware data scoring mechanism. While the components are relatively simple, the overall formulation is neat and practically motivated. However, the method is based on a token-level IRR signal, the paper does not provide much token-level analysis of what the gating mechanism actually prefers or suppresses.

---

> ### Author Rebuttal · Authors · 2026-03-31
>
> Hi, Reviewer Jq1L:
>
> Thank you for recognizing the simplicity, efficiency, and effectiveness of TarGATE. Your main concerns focus on reference-set robustness, IRR aggregation, interpretability, and the selector-scale effect. Unless noted, all added experiments follow the original main setting. We address them below.
>
> ***`Q1: "Sensitivity to reference-set quality and size"`***
>
> **R1:** We added direct robustness tests on GSM8K using Qwen3-0.6B as selector and Qwen3-1.7B as target, while keeping the evaluation split fixed so that only the reference set changes.
>
> |Reference size|50%|100%|150%|200%|
> |---|---:|---:|---:|---:|
> |TarGATE|60.8|61.6|61.8|61.7|
>
> |Noise ratio in reference|0|25%|50%|75%|100%|
> |---|---:|---:|---:|---:|---:|
> |TarGATE|61.6|61.4|60.4|59.7|59.3|
>
> These results support two points. First, TarGATE is **not brittle to moderate reference-size changes**: performance remains nearly unchanged from 100% to 200%, and drops only modestly at 50%. Second, TarGATE shows **expected sensitivity to reference quality**: when we progressively mix unrelated CSQA samples into the GSM8K reference set, performance degrades gradually. This indicates that the method indeed relies on target-relevant signal while retaining considerable robustness when the reference set is imperfect. We agree this is an important issue, and we will explicitly discuss this limitation in the revised paper.
>
> ***`Q2: "Why use global mean IRR as the final score"`***
>
> **R2:** We added IRR aggregation ablations on GSM8K under the same selector/target setting:
>
> |Aggregation|Global|Last-layer|Max-layer|Top-k|Answer-only|High-entropy|
> |---|---:|---:|---:|---:|---:|---:|
> |Score|62.1|60.5|61.7|61.1|54.1|**62.3**|
>
> The results show that **global mean IRR is already near-best**. The only slightly stronger variant is entropy-weighted averaging (62.3 vs 62.1), while all other more specialized choices are weaker, sometimes substantially so (e.g., answer-only averaging). This supports our design judgment: useful target-aware signal is distributed across layers and tokens, so a simple global mean captures it well without introducing task-specific heuristics. Balancing performance, simplicity, and generality, we therefore keep the global mean as the default. At the same time, we agree that aggregation design itself remains a worthwhile research question, and we will expand this discussion in the final version and treat richer IRR aggregation as future work.
>
> ***`Q3: "More analysis of what TarGATE selects"`***
>
> **R3:** We agree that this concern is best answered with distribution-, sample-, and token-level evidence together. The paper already shows that TarGATE shifts selection toward target-relevant categories, including more mathematical reasoning for GSM8K and 94.3% Code Engineering on MBPP. We now provide **Figures 1-3** in the anonymous repo (https://anonymous.4open.science/r/targate-rebuttal-D82B). At the sample level, TarGATE's top GSM8K example is a multi-step arithmetic problem and its top MBPP example is executable Python code, whereas LESS and especially RDS+ drift to weaker or irrelevant patterns. At the token level, higher IRR concentrates on numbers, arithmetic symbols, reasoning steps, and equation-construction cues, while lower-IRR tiers contain more punctuation and formatting fragments. Together, these results suggest TarGATE favors **target-relevant capability rather than superficial overlap alone**. We agree, however, that **a full token-level account remains future work**.
>
> ***`Q4: "Why can the 0.6B selector slightly outperform the 1.7B selector"`***
>
> **R4:** We summarize the selector results on Qwen3-1.7B:
>
> |Selector (TarGATE)|CQA|GSM8K|MBPP|
> |---|---:|---:|---:|
> |Qwen3-0.6B|73.7|**62.1**|**57.2**|
> |Qwen3-1.7B|**74.1**|61.1|56.4|
> |Gemma-3-1B-PT|73.1|61.3|56.8|
> |Llama-3.2-1B|72.6|60.6|56.4|
>
> Together with our original real-world averages (68.2 vs 67.6), these results support the explanation already noted in Sec. 4.2: **smaller selectors can sometimes transfer better**. This is consistent with LESS (Xia et al., 2024), which reports that data selected with a small model remains effective for larger and different-family target models, and argues that useful selection should capture underlying reasoning/skill type rather than surface-form cues. Our new 1B-scale results strengthen the same interpretation because the pattern is not confined to one Qwen pair. Meanwhile, **TarGATE remains relatively robust to selector scale**: even with the 1.7B selector, it still reaches 57.4 on Llama3-8B/GSM8K, well above 45.7 for Grad-Sim. We will make this discussion and citation explicit in the revision.
>
> We will also revise the paper to discuss the three limitations you highlighted: reference quality, token-level analysis, and selector-scale effects.
>
> **Thank you again for your valuable feedback, which has greatly improved our work. We hope these responses address your concerns, and we would be happy to clarify any remaining questions.**

---

> > ### Author Rebuttal · Reviewer_Jq1L · 2026-04-03
> >
> > Thanks for the authors response. My concerns are fully addressed. Yet since many are promised in final revision, I'll raise my score to 4.

---

> > > ### Author Response · Authors · 2026-04-03
> > >
> > > We are deeply grateful for your positive assessment and for the insightful comments that helped strengthen our paper throughout the review process. Thank you once again for your expert guidance and your tremendous support of our research.

---

### Decision · Program_Chairs · 2026-04-30

**Decision:**

Accept (regular)

**Comment:**

This paper proposes TarGATE, a target-aware data selection method for instruction tuning that uses lightweight token-attenuation gates to derive an Information Retention Ratio (IRR) as a data quality signal. The paper’s main strengths are its simple and practical formulation, favorable efficiency relative to gradient-based baselines, and strong empirical results across both synthetic noisy and real-world large-scale settings.

The rebuttal addressed most of the major concerns by providing additional evidence on robustness, interpretability, broader selector/target and task coverage, and comparisons to additional baselines. While some questions remain about broader general-purpose applicability and deeper interpretability, the paper is technically solid and makes a useful contribution. I therefore recommend weak accept.